| Open Peer Review | Bacteriology | Methods and Protocols

# Nanopore adaptive sampling effectively enriches bacterial plasmids

Jens-Uwe Ulrich,[1,2,3] Lennard Epping,[4] Tanja Pilz,[4] Birgit Walther,[5] Kerstin Stingl,[6] Torsten Semmler,[4] Bernhard Y. Renard[1]

**ABSTRACT** Bacterial plasmids play a major role in the spread of antibiotic resistance genes. However, their characterization via DNA sequencing suffers from the low abundance of plasmid DNA in those samples. Although sample preparation methods can enrich the proportion of plasmid DNA before sequencing, these methods are expensive and laborious, and they might introduce a bias by enriching only for specific plasmid DNA sequences. Nanopore adaptive sampling could overcome these issues by rejecting uninteresting DNA molecules during the sequencing process. In this study, we assess the application of adaptive sampling for the enrichment of low-abundant plasmids in known bacterial isolates using two different adaptive sampling tools. We show that a significant enrichment can be achieved even on expired flow cells. By applying adaptive sampling, we also improve the quality of *de novo* plasmid assemblies and reduce the sequencing time. However, our experiments also highlight issues with adaptive sampling if target and non-target sequences span similar regions.

**IMPORTANCE** Antimicrobial resistance causes millions of deaths every year. Mobile genetic elements like bacterial plasmids are key drivers for the dissemination of antimicrobial resistance genes. This makes the characterization of plasmids via DNA sequencing an important tool for clinical microbiologists. Since plasmids are often underrepresented in bacterial samples, plasmid sequencing can be challenging and laborious. To accelerate the sequencing process, we evaluate nanopore adaptive sampling as an *in silico* method for the enrichment of low-abundant plasmids. Our results show the potential of this cost-efficient method for future plasmid research but also indicate issues that arise from using reference sequences.

**KEYWORDS** adaptive sampling, readuntil, nanopore sequencing, plasmid, bacteria, enrichment

Infectious diseases caused by bacterial pathogens have lost their threat to people living in high-income countries due to the discovery of antibiotic drugs within the last 70 years. However, adaptation processes within bacteria cause these drugs to lose their effectiveness in treating infectious diseases. The emergence of such antimicrobial resistance (AMR) already poses a significant threat to public health, with an estimated 4.95 million deaths associated with bacterial AMR in 2019 (1), and will even worsen, with around 10 million expected deaths per year by 2050 (2, 3). Besides vertically passing antimicrobial resistance genes (ARG) to their offspring, bacteria can also transfer ARGs across the bacterial population by horizontal gene transfer. This process is mediated via mobile genetic elements, such as plasmids, which are epichromosomal DNA elements unique to bacteria (4, 5). Plasmids are a major driver in the spread of ARGs in bacterial populations (6) and have recently been found to accelerate bacterial evolution by enhancing the adaptation of the bacterial chromosome (7). Classifying plasmid types is crucial to understanding antibiotic resistance transmission between bacteria. Several

Address correspondence to Jens-Uwe Ulrich, jens-uwe.ulrich@hpi.de, or Bernhard Y. Renard, bernhard.renard@hpi.de.

J.U.U. and B.Y.R. have filed a patent application on selective nanopore sequencing approaches.

See the funding table on p. 14.

recent studies have shown the benefit of whole genome sequencing for classifying plasmid types (8, 9). In particular, the emergence of long-read sequencing by Oxford Nanopore Technologies (ONTs) promises improvements for outbreak investigations due to its lower capital investment and shorter turnaround times (10, 11). However, these methods suffer from the small proportion of plasmid DNA within the sequenced samples (12). Therefore, a large proportion of the plasmids in such samples is probably missed, or the sequencing depth is insufficient to assemble them correctly (13). Thus, additional sample preparation steps are required to isolate or enrich plasmids before DNA sequencing, but they are too expensive and laborious for applications in clinical diagnostic settings. These are particularly interesting for nosocomial infections where the potential pathogens are known, and the focus lies in identifying antibiotic resistance genes, which are mainly present on plasmids and could impact the treatment of patients. While nanopore sequencing has been shown to reconstruct plasmids accurately (14), the technology offers a feature called adaptive sampling (AS) that has the potential to improve plasmid classification. First described in 2016 by Loose et al. (15), nanopore adaptive sampling has been increasingly used for *in silico* target enrichment within the last 2 years. Here, DNA molecules can be rejected from individual nanopores if the corresponding sequence is not interesting for downstream analysis. Pulling out unwanted DNA frees the nanopore for the following molecule to be sequenced and reduces the time spent sequencing uninteresting DNA fragments. Different tools implement adaptive sampling (16–18), using dynamic time warping (UNCALLED), read mapping (Readfish, MinKNOW), or k-mer-based (ReadBouncer) strategies, all performing rejection decisions by analyzing the first 160–450 base pairs (bp) of each read. Recently, deep learning-based tools like SquiggleNet and DeepSelectNet have also been developed, addressing host depletion in human microbiome samples (19, 20). The potential enrichment reached by using adaptive sampling was already shown, and even mathematical models that predict the enrichment factor were recently described by some groups (17, 21, 22). In one study, Marquet et al. (23) could enrich the microbiome in human vaginal samples by depleting host DNA. Further, Kipp et al. (24) used adaptive sampling to enrich bacterial pathogens in tick samples, while Viehweger et al. (22) even enriched single ARGs in human microbiome samples. In the present proof-of-concept study, we investigate the efficiency of adaptive sampling to enrich plasmid sequences in the easiest use case, where the bacterial references are known. All five bacterial organisms we sequenced are known human pathogens that harbor antibiotic resistance genes on their plasmids. For testing whether similar enrichment can be achieved with different tools, we chose to perform adaptive sampling with two tools, which were shown to have high read classification performance, namely, the built-in adaptive sampling feature of MinKNOW (referred to as "MinKNOW" in this manuscript) and ReadBouncer (18, 19). Both tools use a combination of basecalling with ONT's Guppy and read classification on the sequence level. While MinKNOW's adaptive sampling feature is based on the Readfish (17) scripts and uses minimap2 to map read prefixes against a given reference sequence set, ReadBouncer utilizes pseudo-mapping based on k-mers and interleaved Bloom filters for making rejection decisions. We refrained from using UNCALLED because Bao et al. (19) showed that the combination of basecalling and mapping has a higher read classification accuracy than UNCALLED. In order to increase sustainability and reduce sequencing costs, we also investigate whether enrichment of plasmids can be achieved with adaptive sampling on expired flow cells with reduced active pores. Finally, we evaluate the effective plasmid enrichment by comparing it to the predicted enrichment calculated by the mathematical model proposed by Martin et al. (21) and demonstrate the usefulness of adaptive sampling for plasmid assemblies.

## RESULTS

In this study, we present the application of nanopore adaptive sampling on the *in silico* enrichment of plasmids by depleting chromosomal reads during the sequencing of bacterial isolates. Therefore, we sequenced five bacterial strains—*Campylobacter*

*jejuni*, *Campylobacter coli, Salmonella enterica*, *Enterobacter hormaechei,* and *Klebsiella pneumoniae*—on four different flow cells for 24 hours, each separated into an adaptive sampling and a control region. All flow cells, except ReadBouncer1, were used 2–3 months after reaching the manufacturer's recommended storage duration, and throughout the manuscript, we will refer to the flow cells according to the adaptive sampling tool used. All of the chosen bacterial strains are clinically important human pathogens with ARG harboring plasmids, which have already been sequenced in our laboratory. Thus, their chromosomal and plasmid reference sequences were sequenced, assembled, and characterized before conducting this study, which provided us with the necessary ground truth for our data analysis. Four of the five bacterial strains harbor one plasmid with a size between 26 and 157 kb. Only *Enterobacter hormaechei* has four different plasmids with sizes ranging from 6 to 310 kb. The chromosome sizes of the five bacterial strains range from 1.6 to 5.4 Mb. We used these chromosomal references as depletion targets for all adaptive sampling experiments conducted in this study. Further information on the five strains can be found in Table S2. We further investigated the quality of the sequencing runs by looking at the number of active sequencing pores, read lengths, and mean Phred quality scores of reads from the control regions. Although the number of active sequencing pores on expired flow cells is generally below the minimum number of active pores covered by the manufacturer's warranty of 800 pores, we did not recognize a significant effect on read lengths and Phred quality scores. Adaptive sampling also does not significantly impact the read quality or the degradation of active sequencing pores. A more detailed description of the results is provided as an investigation of the sequencing runs in the Supplemental Material.

## Adaptive sampling increases plasmid yield while decreasing overall sequencing yield

We analyzed the effect of adaptive sampling on overall sequencing yield and the number of sequenced reads. In Fig. 1a and b, we see that for all four flow cells, the sequencing yield on adaptive sampling regions is significantly reduced in comparison to control regions. This observation aligns with previous studies (17, 21) and originates mainly from fewer active sequencing channels in adaptive sampling regions and also from a reduced overall time spent for sequencing the DNA and more overall time needed to capture the DNA molecules when adaptive sampling is applied. Assuming a read capturing time of 0.5 seconds and a sequencing pace of 420 bp/second, the second point can account for up to 50 Mbp if 250,000 additional reads are sequenced in the adaptive sampling region. However, this explains only a small fraction of the reduced yield, showing that fewer active sequencing channels are the main driver for the reduced overall yield. In general, we increased the yield in sequenced plasmid base pairs with adaptive sampling for all but one bacterial sample (Fig. 1b and 2). In this context, we also see on all four flow cells a higher number of reads sequenced on the adaptive sampling regions than on the control regions (Fig. 1c and d). Thus, many reads are classified as chromosomal by the adaptive sampling tools and rejected from the pores, leading to more reads sequenced on the adaptive sampling regions. Here, the flow cell run ReadBouncer2 has a higher number of reads on the adaptive sampling region than ReadBouncer1. This results from a lower relative plasmid abundance in samples sequenced on flow cell ReadBouncer2, which leads to a larger number of chromosomal reads that were rejected on the adaptive sampling region of flow cell ReadBouncer2 (approx. 400,000) than on the adaptive sampling region of ReadBouncer1 (approx. 370,000).

## Rejecting chromosomal reads increases the relative plasmid abundance

In our four experiments, we investigate the potential relative enrichment of plasmid sequences in bacterial samples by rejecting the chromosomal reads using adaptive sampling. First, we determined the percentage of sequenced chromosomal and plasmid base pairs for each sample from the adaptive sampling and control regions. Figure 3 presents each sample's chromosome and plasmid base pair percentages on control and

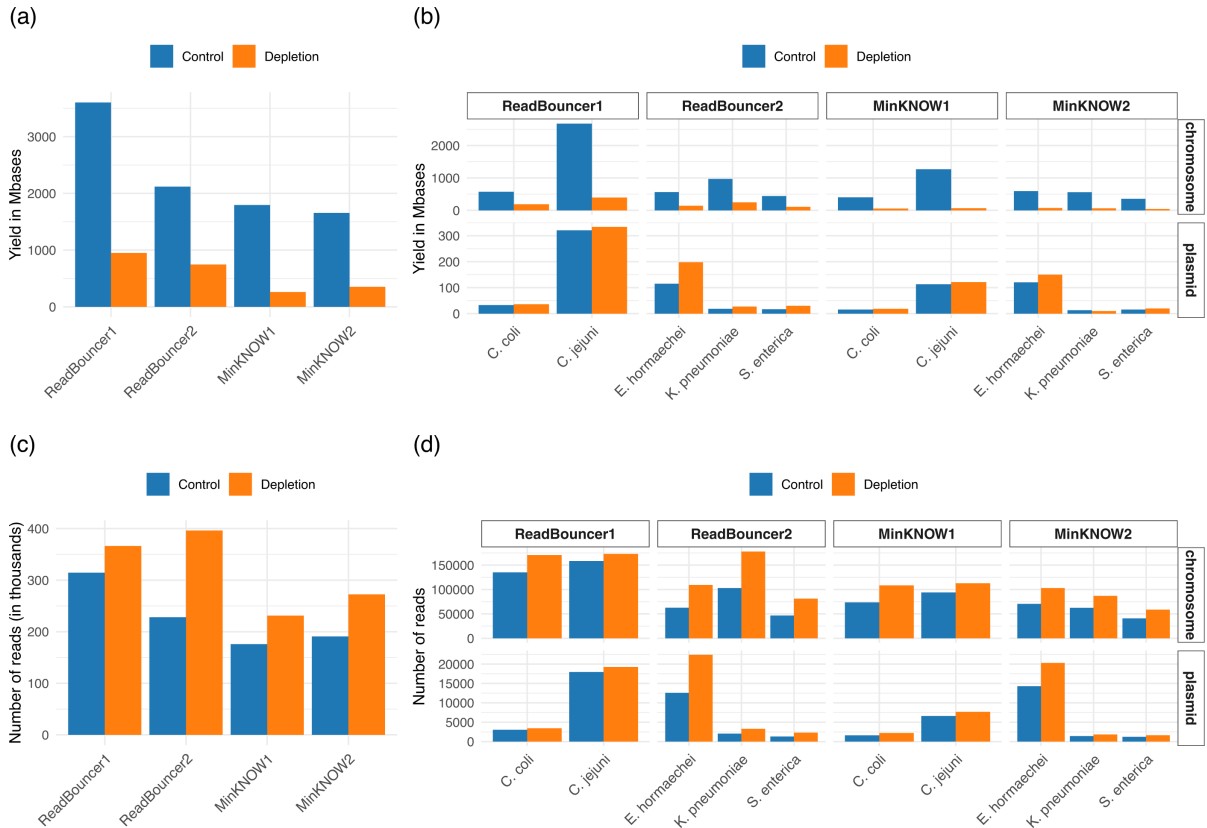

**FIG 1** Comparison of flow cell yield in terms of sequenced base pairs and reads after 24 hours. (a) Yield in megabases for each flow cell separated by control and adaptive sampling region (depletion). (b) Yield in megabases for each flow cell region separated by plasmid and chromosome. (c) Number of sequenced reads for each flow cell separated by control and adaptive sampling region (depletion). (d) Number of sequenced reads for each flow cell region separated by plasmid and chromosome.

adaptive sampling regions. After 24 hours of sequencing, we see that adaptive sampling increases the relative abundance of plasmid base pairs for all samples on the four flow cells. For instance, we could increase the abundance of *Campylobacter coli* plasmid bases from 3.68% to 24.75% when rejecting chromosomal reads with MinKNOW. We further examined whether the relative plasmid enrichment by composition and yield we observe in our experiments corresponds to the predicted relative compositional enrichment by the mathematical model proposed by Martin et al. (21). Therefore, we calculated the relative compositional enrichment by dividing the percentage of plasmid base pairs from adaptive sampling regions by those from control regions. Accordingly, we calculated the relative enrichment by yield using the number of sequenced plasmid base pairs from adaptive sampling and control regions. As predicted by the model, the enrichment factor was higher for samples with less abundant plasmids (Fig. 4a). The highest levels of compositional enrichment were obtained using MinKNOW, which can be explained by faster rejection decisions. An analysis of rejected reads revealed shorter read lengths for MinKNOW compared to ReadBouncer, which is caused by rejection decisions based on short read prefixes (see Fig. S9). In the histogram, we see that reads rejected by ReadBouncer are longer than those rejected by MinKNOW, with an average length of 848 bp compared to 520 bp. This confirmed our assumption that ReadBouncer rejects reads later during the adaptive sampling process, resulting in a higher abundance of unwanted chromosomal base pairs in the final output. To avoid confusion, we have to note that the lengths of rejected reads in the final output are not the same as the read prefix (or chunk) length used by adaptive sampling tools for making rejection decisions. Lengths of rejected reads in the final output represent the time needed for the whole decision process, including time for communication with the API and mapping of reads

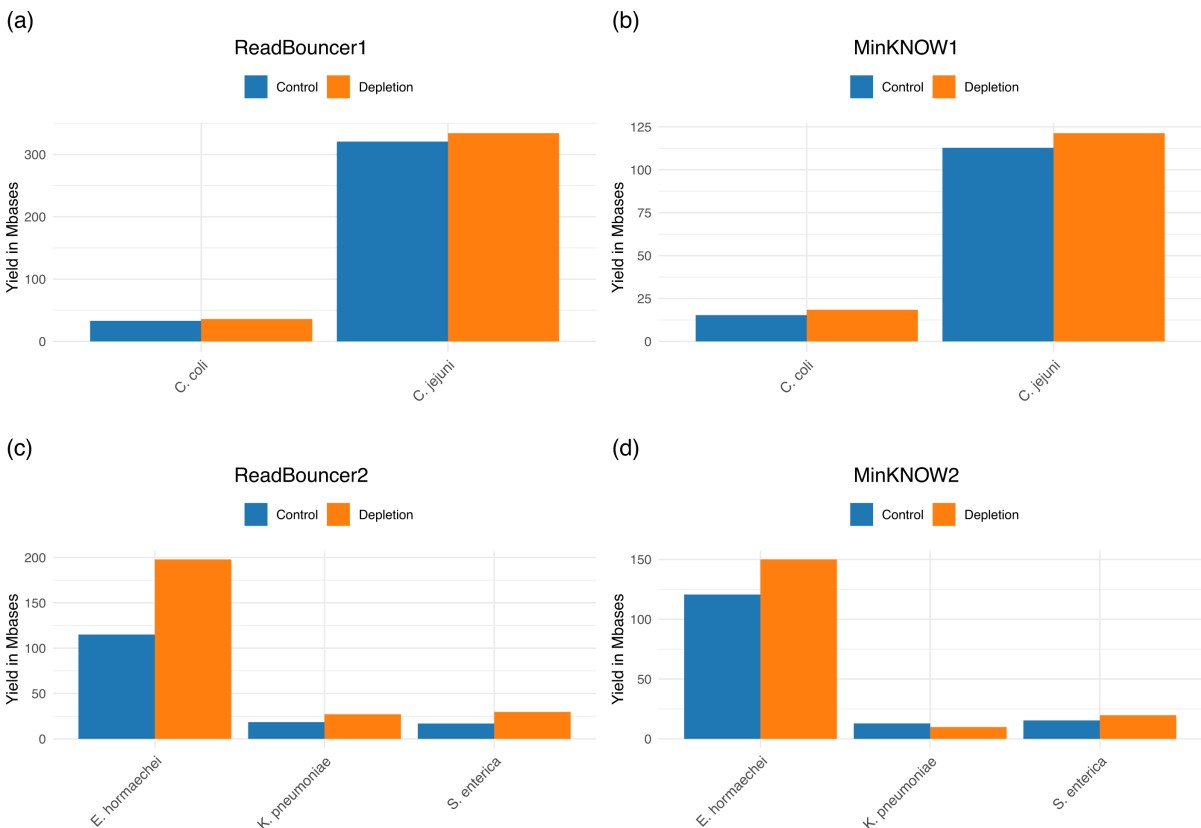

**FIG 2** Comparison of plasmid yield in megabases for each flow cell regarding sequenced base pairs after 24 hours. There is a small increase in plasmid yield for the two *Campylobacter* samples from (a) ReadBouncer1 and (b) MinKNOW1. (c) Plasmid yield is increased for all three bacterial samples from ReadBouncer2. (d) Plasmid yield is increased for two of the three samples from flow cell MinKNOW2. There is a decreased plasmid yield for *K. pneumoniae* with adaptive sampling using MinKNOW.

against index data structures. The predictions from the mathematical model by Martin et al. (21) correlated moderately with our observations (Pearson's $r = 0.55$), as shown in Fig. 4b. In contrast, the original relative plasmid abundance has no impact on the relative enrichment by yield (Fig. 4), with enrichment by yield being significantly less than enrichment by composition. When using adaptive sampling, the composition of a sample changes, e.g., from 90% chromosome/10% plasmid to 80% plasmid/20% chromosome. Here, we see that the lower the plasmid abundance was in the original sample, the higher the fold change for this compositional abundance. However, when we use adaptive sampling, plasmid abundance does not impact the fold change in sequenced plasmid bases. Irrespective of whether we had 5% or 10% plasmid bases in our sample, the relative enrichment in plasmid bases will be between 1.1 and 1.8. Finally, we also noticed that the predicted enrichment values by the model do not correlate with the observed enrichment values by yield (Pearson's $r = -0.07$, Fig. 4d). Thus, the model is able to predict the relative compositional enrichment but fails to predict the relative enrichment by yield.

## Effective enrichment of plasmids by yield, read number, and mean depth of coverage

We examine the effective relative plasmid enrichment at different time points of sequencing for each experiment by calculating the plasmid enrichment for the five bacterial species in 30-minute intervals. According to equation 1 defined in Materials and Methods, the enrichment by yield is the ratio of cumulative plasmid bases from the adaptive sampling region and the control region at time point *t*. We calculate the

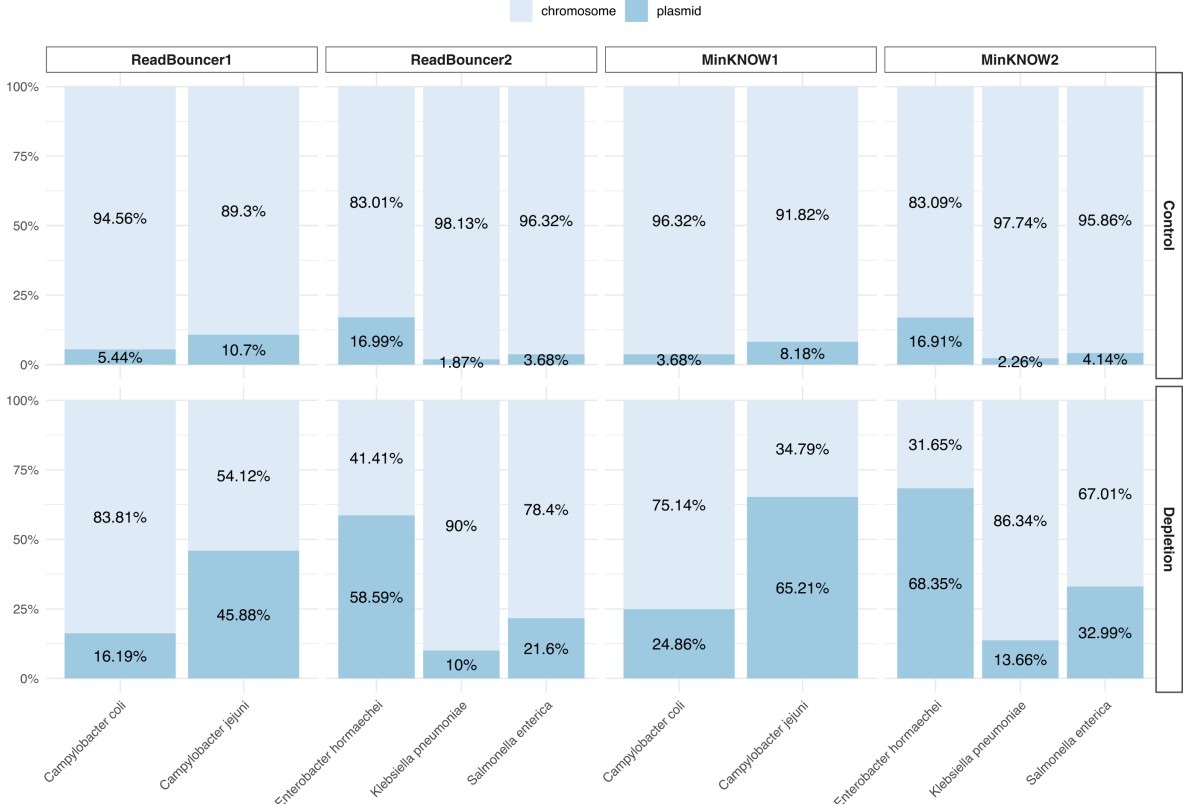

**FIG 3** Comparison of relative plasmid abundances in five bacterial samples. Adaptive sampling with MinKNOW was used on flow cells MinKNOW1 and MinKNOW2, and ReadBouncer was used as an adaptive sampling tool on flow cells ReadBouncer1 and ReadBouncer2. For all experiments, plasmid abundances for each sample were measured after 24 hours of sequencing for control regions and adaptive sampling regions (depletion). Plasmid abundances are highest when using MinKNOW for the depletion of chromosomal nanopore reads.

enrichment by the number of plasmid reads and mean depth of coverage in the same manner as proposed by equations 2 and 3 outlined in Data Analysis. In general, we see a steep increase in enrichment at the beginning of each experiment, followed by a steady decline as the experiments progress. While the steep increase at the beginning is very surprising, we assume that the slow decrease is caused by pore degradation and most chromosomal reads being rejected early in the experiments, resulting in a relatively constant number of plasmid reads sequenced throughout the later stages in the experiments on both sides of the flow cells. Figure 5a illustrates that we obtain an enrichment of plasmid reads for all samples in all experiments at any given time point. This observation confirms that the number of sequenced plasmid reads can be increased by using adaptive sampling. We can see the same effect for the enrichment by yield (Fig. 5b) for all but one sample. For the *Klebsiella pneumoniae* sample of flow cell MinKNOW2, we observe that the number of plasmid bases from adaptive sampling is less than that from the control channels. Thus, we failed to obtain an enrichment of *Klebsiella pneumoniae* plasmids in that experiment where we used MinKNOW to deplete chromosomal reads. For all other samples, we observe an enrichment of 1.1×–1.8× after 24 hours, corresponding to 10%–80% more plasmid data when using adaptive sampling, even when using expired flow cells with reduced active pores. We further investigated the difference in enrichment between the same samples from experiments ReadBouncer2 and MinKNOW2. First, flow cell MinNKOW2 has fewer active sequencing channels (see Fig. S1) and yields less sequenced base pairs than the flow cell from experiment Read-Bouncer2 (see Fig. 1). Figure S7 also illustrates that the average read quality in the adaptive sampling region of flow cell MinKNOW2 is smaller than that for flow cell ReadBouncer2. Both observations suggest a decreased pore quality of flow cell

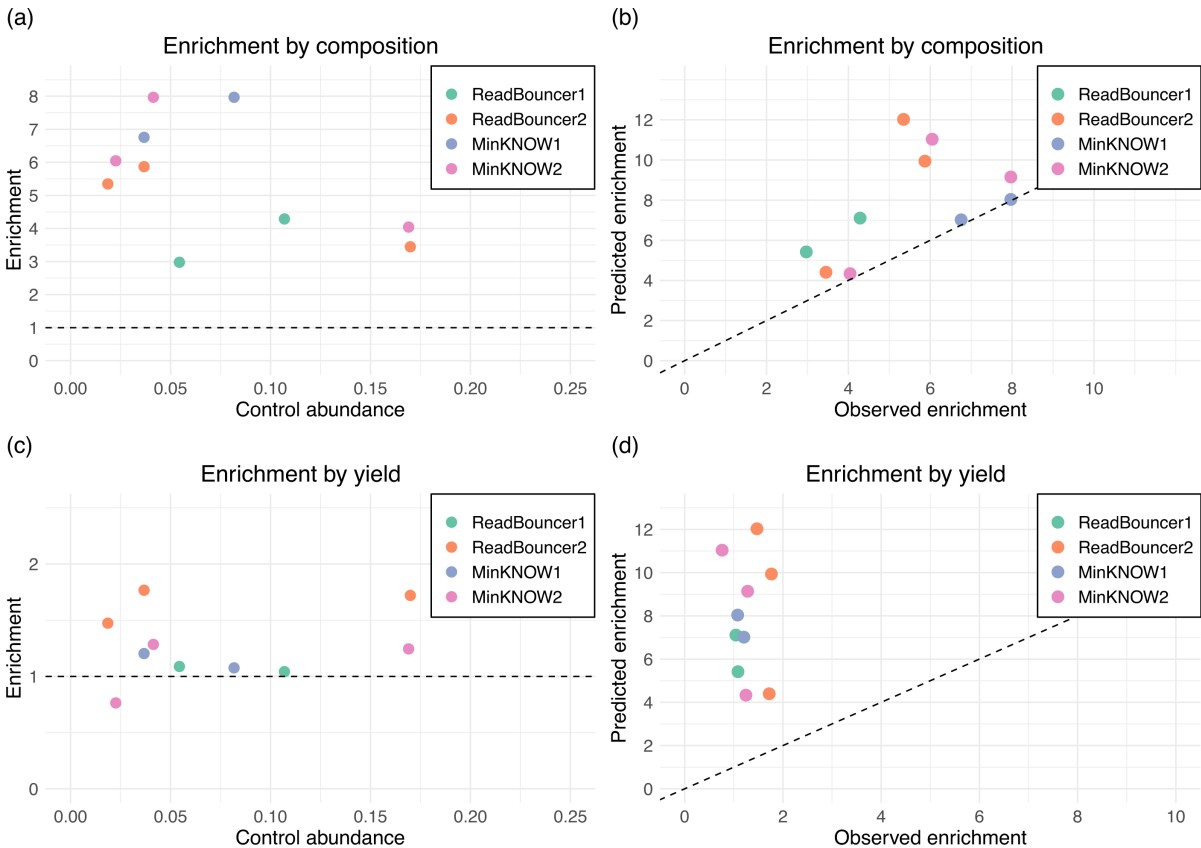

**FIG 4** Scatterplots for relative plasmid enrichment by composition and yield. (a) Observed enrichment factor by composition against relative abundance. Each point represents a bacterial sample, with the position on the *x*-axis indicating the original relative abundance of plasmids in the sample and the position on the *y*-axis indicating the enrichment factor obtained. Points above the dashed line indicate enrichment, and points below the line indicate plasmid sequence depletion. (b) Correlation between observed enrichment values by composition and predicted enrichment values by the mathematical model (Pearson's *r* of 0.55). (c) Enrichment factor by yield against relative abundance. Relative enrichment of plasmids is independent of the plasmid abundance in the sample. (d) Correlation between observed enrichment values by yield and predicted enrichment values by the mathematical model (Pearson's *r* of −0.07). The model fails to predict relative plasmid enrichment by yield.

MinKNOW2. Although this might explain the reduced enrichment by yield in this experiment, it does not explain why there is an effective depletion of plasmid bases for *Klebsiella pneumoniae* in experiment MinKNOW2. Thus, we identified all reads from the final output that mapped against the *Klebsiella pneumoniae* plasmids but were rejected by MinKNOW. We extracted these falsely rejected plasmid reads of *Klebsiella pneumoniae* and mapped them to the corresponding bacterial chromosome reference sequences with minimap2. Using samtools depth, we could identify four regions (between 829 and 2,101 bp long) on the *Klebsiella pneumoniae* chromosome with read depth ≥10. An investigation of the annotated GenBank file revealed that two of those regions code for *IS6-like element IS26 family transposase* and *IS110-like element IS5075 family transposase*, both belonging to the group of insertion sequences (ISs), which are small DNA segments (<2 kbp) that encode an enzyme, the transposase, which catalyzes the DNA cleavage and strand transfer reactions enabling movement of the element between DNA molecules (25). The third region codes for *group II intron reverse transcriptase/maturase*, a mobile genetic element encoding reverse transcriptases that are important for RNA splicing (maturase activity) by helping the intron RNA fold into the catalytically active structure (26), and the fourth region encodes *CusA/CzcA family heavy metal efflux RND*, which is an efflux pump transporting heavy metal ions out of the bacterial cell and is important for antimicrobial resistance (27). These findings reveal regions of high identity between *Klebsiella pneumoniae* plasmid targets and non-target chromosome sequences. Such

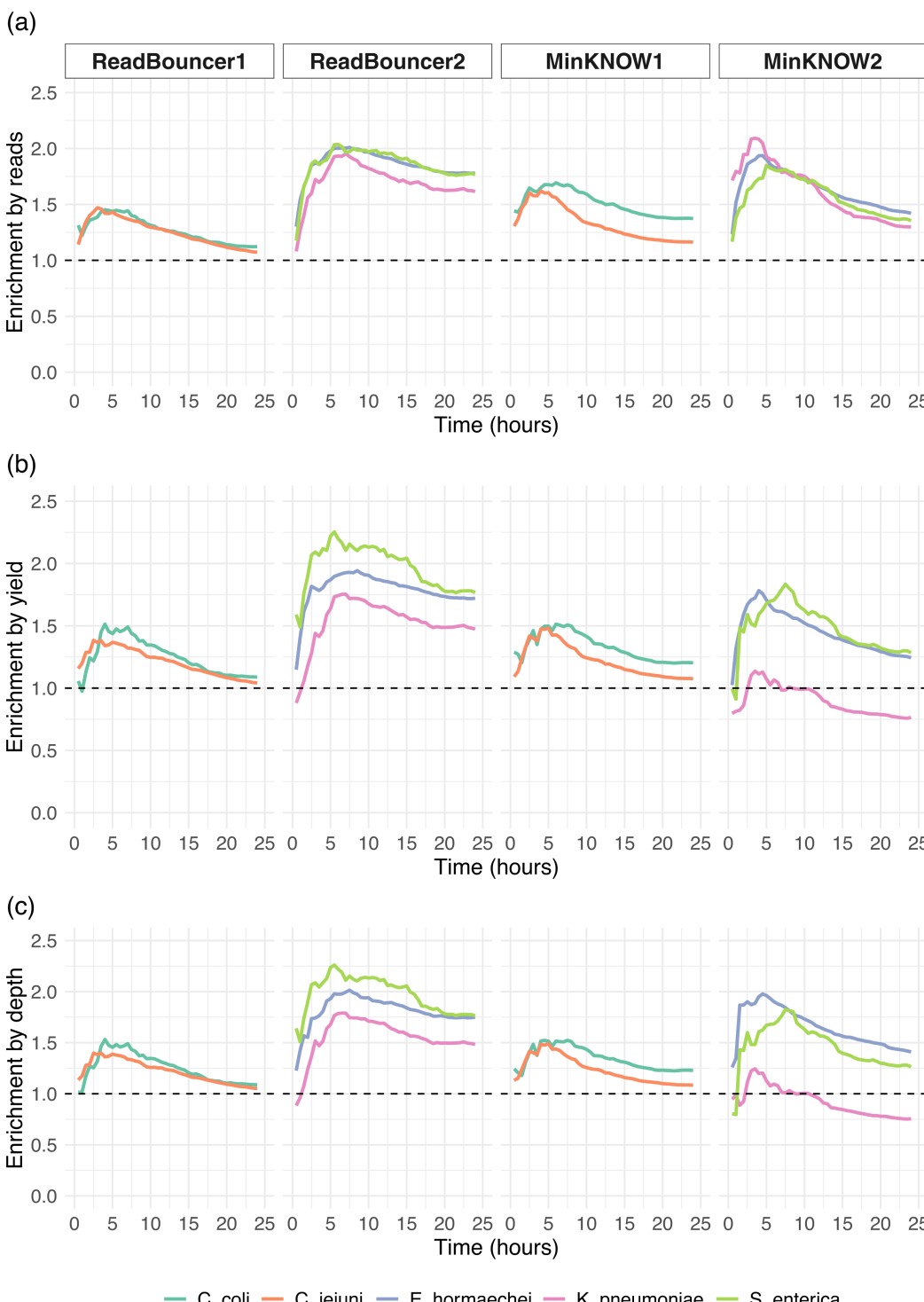

**FIG 5** Comparison of enrichment in five bacterial samples. (a) Enrichment by the number of plasmid reads for the five bacterial strains across all four sequencing runs. (b) Enrichment by the number of sequenced plasmid bases for all five bacterial strains across the four sequencing runs. (c) Enrichment by mean depth of coverage of plasmid references for the five bacterial strains across the four sequencing runs. The dashed line indicates the enrichment factor threshold, which values above 1.0, implying effective enrichment. Values below 1.0 imply depletion of plasmid sequences. All strains but the *Klebsiella pneumoniae* sample show a slight enrichment, where MinKNOW was used for adaptive sampling.

similar regions between target and non-target sequences pose a challenge for the application of nanopore adaptive sampling and potentially lead to an increased number of falsely rejected target reads. Here, it seems that ReadBouncer can avoid a high number of false rejections by using longer read prefixes (see Fig. S7 and S9) for making rejection decisions. Our observations suggest that using more sequence information by increasing the chunk size for adaptive sampling with MinKNOW could circumvent such issues, which is currently not a tunable parameter.

## Adaptive sampling helps improve plasmid assemblies

Our experiments demonstrated an effective enrichment of plasmids after 2–5 hours by using adaptive sampling. Since plasmid assemblies are possible after 3–4 hours of sequencing without adaptive sampling (10), we wanted to see if adaptive sampling enables faster plasmid assemblies. In order to evaluate the effect of adaptive sampling on the *de novo* assembly of low-abundant plasmids, we separately assembled all available reads from the control and adaptive sampling regions after 1, 2, and 3 hours of sequencing using metaFlye assembler (28). After one round of polishing with Medaka consensus, we measured quality metrics for the final assemblies using Quast (29). Table S3 shows results for all bacterial isolates from each sequencing run. In almost all cases, we observe a higher average depth of coverage for plasmids sequenced with adaptive sampling. Only for *Klebsiella pneumoniae*, plasmid assemblies with adaptive sampling show no benefit over control assemblies after 1 and 2 hours of sequencing. However, after 3 hours, we see a slightly increased average coverage depth and fewer mismatches and indels per 100 kbp. Further, after 2 and 3 hours, only *de novo* assemblies of adaptive sampling regions from ReadBouncer correctly assemble the plasmid into one contig. We also recognize that metaFlye only assembles all four plasmids from *Enterobacter hormaechei* at all three time points for data from the adaptive sampling region of the ReadBouncer run. Although we effectively enrich plasmid sequences for four out of five bacterial isolates, we cannot consistently demonstrate an improvement with regard to the number of mismatches and indels per 100 kbp. For some assemblies, the results also seem counterintuitive, with increasing mismatches and indels while the reference coverage increases too. This can be observed particularly for *Campylobacter coli* assemblies. Since we do not see these results across all assemblies from adaptive and control regions, we suggest that these results are artifacts from the Quast pipeline. In general, our results show that adaptive sampling can improve the quality of plasmid assemblies by increasing the depth of coverage even when sequencing was performed on flow cells with fewer active pores.

## DISCUSSION

Recent studies have demonstrated the utility of adaptive sampling for the enrichment of underrepresented sequences in various applications, such as host depletion in human vaginal samples or antibiotic resistance gene enrichment in metagenomics samples. In this study, we examine the potential of adaptive sampling for the enrichment of low-abundant plasmid sequences by rejecting chromosomal sequences in bacterial isolate samples. We demonstrate the possibility of using even older or expired flow cells with fewer active sequencing pores for the *in silico* enrichment via adaptive sampling. Since we wanted to know if enrichment is independent of the adaptive sampling tool, we evaluated plasmid enrichment for two tools, namely, ReadBouncer and ONT's MinKNOW sequencing control software. Our study was by no means designed to benchmark different adaptive sampling tools, which would require the inclusion of more tools and a setup that ensures that all tools use the same amount of sequence information for making rejection decisions. This can only be ensured by using adaptive sampling simulation tools like Icarust (30) or SimReadUntil (31). However, both tools consistently enriched low-abundant plasmid sequences, with only one exception where MinKNOW failed to enrich plasmid sequences for *Klebsiella pneumoniae*. The fact that ReadBouncer uses longer read prefixes for the decision-making algorithm seems to

prevent false rejection decisions. Unfortunately, the prefix length, alignment identity, or minimum alignment length for decision-making cannot be parameterized via MinKNOW, which suggests that more tunable tools such as ReadBouncer are better suited for complex samples. However, MinKNOW is more user-friendly for less complex samples and potentially achieves higher enrichment values by faster rejection decisions. The enrichment by yield, the most critical value for researchers, lies for all but one sample in our experiments between 1.1× and 1.8× after 24 hours of sequencing on an ONT MinION sequencing device. We also demonstrated that the difference between enrichment by yield, number of reads, and mean depth of coverage is negligible in all our samples. *De novo* assemblies of plasmids are possible within 2 hours of sequencing with adaptive sampling and show even better results than plasmid assemblies without adaptive sampling. These results reflect the benefit of adaptive sampling in assembling low-abundant plasmid sequences. Since we sequenced three bacterial isolates on only half an expired flow cell, we reason that up to 20 bacterial isolates can be sequenced on a flow cell with adaptive sampling for plasmid enrichment. Our experiments showed that expired flow cells (2–3 months after the manufacturer's recommended storage time) with decreased active pores could be used in combination with adaptive sampling. Previous studies demonstrated that the number of active sequencing pores decreases faster when using adaptive sampling. Although we observe that the number of active sequencing channels in control regions is higher than that in the adaptive sampling region, we cannot see a faster deterioration of pores in our study. We also do not see a negative impact of adaptive sampling on the enrichment of target sequences and the average quality of sequenced reads. Thus, we encourage researchers to use flow cells with reduced active pores in adaptive sampling experiments for more sustainable lab experiments and cost savings in core facilities and larger research institutions. Our results show that rejecting chromosomal sequences with adaptive sampling increases the abundance of plasmid sequences in the final output. Depending on the plasmid abundance in the original sample, the values for plasmid enrichment by composition are between 2.5× and 8×. These observations moderately correlate with the predictions from the mathematical model proposed by Martin et al. (21). Furthermore, a consistent enrichment of plasmid sequences with regard to the number of base pairs, number of reads, and depth of coverage was shown by using adaptive sampling. Independent of the size of the sequencing libraries, we could increase the amount of sequenced plasmid base pairs by 10%–80% after 24 hours of sequencing. However, in one experiment, we recognized the depletion of plasmid sequences of *Klebsiella pneumoniae* after 24 hours when ONT's MinKNOW was used as an adaptive sampling tool. Our investigations reveal that regions with high sequence identity located both on the chromosome and the plasmid lead to false read rejections, which result in a depletion of the targeted plasmid sequences. This highlights potential issues with the usage of nanopore adaptive sampling and sounds a note of caution if target and non-target sequences are similar. We hypothesize from our findings that using larger read chunks for making rejection decisions could circumvent this issue. However, such an examination is beyond the scope of this study and needs systematic investigations to find the optimal read chunk length that minimizes false rejection decisions while still rejecting unwanted reads fast enough to obtain sufficient enrichment. Both adaptive sampling tools used in this study need known reference sequences to reject the chromosomal reads. If the bacterial species in the given sample are unknown, a more extensive reference database of all potential bacterial chromosome references must be used to enrich plasmids successfully. Alternatively, researchers could also do a targeted enrichment of the plasmids by using plasmid databases such as PLSDB (32) and reject all reads that do not match the database. However, this approach risks missing unknown plasmids not covered by the database. Using specific plasmid markers, like the origin of replication, to classify unknown plasmids correctly is also tricky in an adaptive sampling experiment. The specific markers would need to be located on the first 1,000 bp of the read to prevent false rejection of plasmid reads. These limitations reinforce the need for improved

classification algorithms that can even classify reads from unknown plasmids based on the raw nanopore signals. We envisage several applications for the *in silico* enrichment of plasmids in the near future. One possibility is the surveillance of plasmid outbreaks in hospital settings. Here, clinicians are interested in studying the transmission of specific antibiotic resistance genes harboring plasmids from one bacterial species to another. Such community transmissions can indicate the selection pressure on bacteria caused by antibiotic pharmaceuticals and help decide on the future usage of the corresponding drugs. Another possible application of adaptive sampling is the improvement of known bacterial assemblies. In this study, we demonstrated the improved time to assembly of plasmids by depleting the known bacterial chromosomes. We plan to develop a pipeline for the real-time *de novo* assembly of bacterial isolates in the future. Using adaptive sampling, we could reject reads that cover assembled regions with a minimum depth of coverage, enriching for unseen or assembled regions with low sequencing depth. In such a way, we could complement the dynamic re-sequencing framework BOSS-RUNS (33) with a dynamic *de novo* adaptive sampling framework. We believe this could improve both the quality of bacterial and plasmid assemblies as well as metagenomics assemblies of unknown bacterial species.

## MATERIALS AND METHODS

### Culture and DNA extraction

*Campylobacter* strains were streaked on Columbia Blood agar (Oxoid, Thermo Fisher Scientific, USA) and incubated at 42°C under a microaerobic atmosphere. The *Enterobacter, Salmonella,* and *Klebsiella* strains used in this study were streaked out on a Luria–Bertani plate and incubated overnight at 37°C. DNA extraction for *Campylobacter jejuni* (GCF_008386335.1) (34) was done using the MagAttract HMW Genomic Extraction Kit (Qiagen). For *Salmonella enterica* (GCA_025839605.1), *Campylobacter coli* (GCF_025908295.1) (35), *Klebsiella pneumoniae IMT44613* (GCF_025837075.1), and *Enterobacter hormaechei IMT49658-3* (GCF_001729785.1) DNA was extracted using the QIAamp DNA Mini Kit (Qiagen, Hilden, Germany) according to the manufacturer's instruction. The total amount of DNA was quantified using a Qubit fluorometer (Thermo Fisher Scientific) and frozen at −80°C until further analysis.

### Library preparation and sequencing

Sample preparation was performed according to the manufacturer's instructions without any optional pre-enrichment steps or size selection using the Rapid Barcoding Kit SQK-RBK004. Different barcodes were used for each of the bacterial isolate samples to correctly assign sequenced reads in the data analysis. Since we used expired flow cells with less expected overall sequencing yield, we decided to sequence only two or three bacterial isolates on one flow cell. Finally, the barcoded samples were sequenced on an Oxford Nanopore MinION (Oxford, UK) using FLO-MIN106D (R9.4.1) flow cells. All sequencing experiments were started via ONT's MinKNOW control software (version 4.5.0).

### *In silico* enrichment via adaptive sampling

We performed four sequencing runs using MinKNOW software v4.5.0 on an Nvidia Jetson AGX Xavier (512-core NVIDIA Volta GPU, 32 GB LPDDR4X Memory) for 24 hours. In all experiments, we compared adaptive sampling with standard sequencing by dividing the flow cells into two parts: adaptive sampling was performed on the first 256 channels, and standard sequencing was performed on channels 257–512. We used a new flow cell with 1,153 active pores for the first run (ReadBouncer1) and sequenced two *Campylobacter* isolates using barcodes RBK01 and RBK02. For the second run (ReadBouncer2), we used an expired flow cell with only 636 active pores for sequencing the three barcoded bacterial isolates (*Enterobacter, Salmonella,* and *Klebsiella*) using barcodes RBK03, RBK04,

and RBK05. The third run (MinKNOW1) used the same *Campylobacter* samples as the first, but we performed sequencing on an expired flow cell with only 557 active pores. For the fourth run (MinKNOW2), we used the identical three bacterial isolates as for the second run and performed sequencing on an expired flow cell with only 718 active pores after the initial flow cell check. On the first two flow cells, we performed adaptive sampling with ReadBouncer (18) using the chromosomal references of the bacterial isolates as depletion targets. Here, a k-mer size of 15, a chunk length of 250 bp, a fragment size of 200,000 bp, and an expected error rate of 5% were used as parameters for the read classification. ReadBouncer's k-mer size parameter was chosen accordingly to the default k-mer size used for mapping with minimap2 (36), which is used by MinKNOW's adaptive sampling feature. The expected error rate reflects the current average per-read accuracy by ONT's Guppy basecaller. The other two parameters are default parameters. For flow cells three and four, MinKNOW's adaptive sampling feature was used, which is based on the Readfish (17) scripts and uses minimap2 (36) to map read prefixes against a given reference sequence set for read classification. We built a minimap2 index file (parameter -x map-ont) for these experiments, including the chromosomal reference sequences, which we used as depletion targets for adaptive sampling. Read prefixes classified as "chromosomal" were rejected from the pore, and decisions were written to log files by both tools, ReadBouncer and MinKNOW. In all experiments, ReadBouncer and MinKNOW used Guppy GPU basecaller (fast model, v6.0.6. Oxford Nanopore Technologies) for real-time basecalling of the raw signal data received from the device. We set the *break_reads_after_seconds* parameter to 0.4, which results in receiving the first chunks of raw data from a read after 0.4 seconds. Both methods concatenate basecalled read chunks to longer prefixes if the prefixes are too short to reliably classify them as a plasmid or chromosome.

## Data analysis

All data analysis scripts were written in Python and R and are freely available in the GitHub repository https://github.com/JensUweUlrich/PlasmidEnrichmentScripts. All plots were created in R using ggplot2. After the sequencing runs were finished, we basecalled and demultiplexed all raw data with Guppy GPU basecaller (super accuracy model, v6.0.6. Oxford Nanopore Technologies). Guppy trimmed barcodes and adapter sequences from the resulting nanopore reads during that process. Afterward, we computed read length metrics (see Table S1; Fig. S1c and S2 to S5) and created contour plots (see Fig. S6 and S7) using the sequencing_summary files provided with the MinKNOW and Guppy output directories. Next, we mapped all demultiplexed and base-called reads against the reference genomes (including plasmid sequences) of the five bacterial strains using minimap2 v2.19 (36) with parameter -x map-ont. Based on the mapping results, we could assign each mapped read to either the bacterial chromosome or plasmid(s) of one of the bacterial isolates to create Fig. 1. We also used the mapping results to calculate the percentage of sequenced plasmid and chromosome base pairs after 24 hours for each bacterial sample, resulting in Fig. 3. We further used the sequencing summary file to separate the reads by their species of origin and partitioned them to comprise the cumulative data from the beginning of each experiment up to 24 hours, separated by 30 minutes of sequencing, which resulted in 48 individual time point data sets. With this information, we calculated for each experiment time point $t$ the plasmid enrichment by yield for each bacterial strain:

$$\text{Enrichment}_{\text{yield}}(t) = \frac{\text{yield}_{\text{AS}}(\text{plasmid}, t)}{\text{yield}_{\text{CTRL}}(\text{plasmid}, t)} \tag{1}$$

where $\text{yield}_{\text{AS}}(\text{plasmid}, t)$ is the number of sequenced plasmid bases of a strain from the adaptive sampling region at time point $t$ and $\text{yield}_{\text{CTRL}}(\text{plasmid}, t)$ is the number of sequenced plasmid bases of a strain from the control region (without adaptive sampling) at time point $t$. Similarly, we calculated the enrichment by the number of reads:

$$\text{Enrichment}_{\text{reads}}(t) = \frac{\text{reads}_{\text{AS}}(\text{plasmid}, t)}{\text{reads}_{\text{CTRL}}(\text{plasmid}, t)} \tag{2}$$

and the enrichment by the mean depth of coverage of the plasmid reference sequences.

$$\text{Enrichment}_{\text{depth}}(t) = \frac{\text{depth}_{\text{AS}}(\text{plasmid}, t)}{\text{depth}_{\text{CTRL}}(\text{plasmid}, t)} \tag{3}$$

According to the definitions above, $\text{reads}_{\text{AS}}(\text{plasmid}, t)$ and $\text{reads}_{\text{CTRL}}(\text{plasmid}, t)$ represent the number of reads from the AS or control region (CTRL) that map to a plasmid of a given bacterial strain at experiment time point $t$. Furthermore, $\text{depth}_{\text{AS}}(\text{plasmid}, t)$ denotes the mean sequencing depth of plasmids from a strain using mapping data from the adaptive sampling region at time point $t$, and $\text{depth}_{\text{CTRL}}(\text{plasmid}, t)$ is the mean sequencing depth of plasmids on the control region at time point $t$. Here, we used samtools coverage (37) to calculate the mean depth of coverage of every species' plasmid reference at each time point for the control and adaptive sampling regions. The different enrichment factor values calculated for each bacterial sample at any of the 48 time points are plotted and shown in Fig. 5. For the plots of active channels over time (Fig. S8), a channel was defined as active from the beginning of the experiment up until the time it sequenced its final molecule (as long as it sequenced at least one molecule). The enrichment by composition shown in Fig. 4a and b was calculated by dividing the relative plasmid abundance from adaptive sampling regions by the relative plasmid abundance from control regions, both shown in Fig. 3. We compared observed enrichment by composition and yield against predicted enrichment values using the mathematical model from Martin et al. (21). To calculate predicted enrichment values, we used the recommended sequencing speed of 420 bp/sec, capture time of 0.5 seconds, decision time of 1 second, mean read lengths for each bacterial sample as provided in Table S1, and plasmid abundances of control regions for each sample as shown in Fig. 3. Since we expect plasmid sequences in our use case scenario to be usually unknown, we also did a *de novo* assembly of the demultiplexed fastq files, containing all reads sequenced after 1 and 2 hours of sequencing. This helps us to estimate the time required to obtain high-quality plasmid assemblies. Therefore, we assembled all demultiplexed nanopore reads from control and adaptive sampling regions separately. Since most long-read assemblers struggle to correctly assemble small plasmids (38), we decided to use Flye/metaFlye assembler (v2.9.2, parameter "--meta") (28, 39), which is meant to improve the assembly of contigs with uneven sequence depths—a situation often experienced with plasmid sequences that are present at high copy numbers in a single cell. Then, we polished the obtained metaFlye assemblies with one round of Medaka consensus (v.1.8.0, default parameters, model r941_min_sup_g507, Oxford Nanopore Technologies) using the same nanopore read set. We assessed the quality of the final assemblies with Quast (v5.2.0) (29) and combined reported metrics like the mean depth of coverage for both time points.

## ACKNOWLEDGMENTS

The authors thank Martin Hölzer, Matthew Huska, and Aleksandar Radonic (RKI) for valuable discussions and comments on the usage of nanopore adaptive sampling. We thank the Genome Sequencing Unit at Robert Koch Institute for sequencing the bacterial strains.

J.U.U. analyzed and interpreted the data and wrote the manuscript. L.E. interpreted the data and wrote the manuscript. T.P. prepared samples and performed the sequencing. B.W. and K.S. performed DNA extraction of the samples. B.Y.R. and T.S. conceived and supervised the study. All authors read and approved the final manuscript.

This work was funded by the German Federal Ministry of Education and Research (BMBF) in the Computational Life Science program (Live-DREAM, 031L0175B) and Global

Health program (ZooSeq, 01KI1905D) and has been supported by a grant from the BMBF/German Center for Infection Research (TI 06.904-FP2019 to B.Y.R.). L.E. was funded by the Deutsche Forschungsgemeinschaft (DFG, German Research Foundation)—Project number 425959793. Open access publishing was also funded by the Deutsche Forschungsgemeinschaft (DFG, German Research Foundation)—Project number 491466077

## AUTHOR AFFILIATIONS

[1]Hasso Plattner Institute, Digital Engineering Faculty, University of Potsdam, Potsdam, Germany

[2]Department of Mathematics and Computer Science, Free University of Berlin, Berlin, Germany

[3]Phylogenomics Unit, Center for Artificial Intelligence in Public Health Research, Robert Koch Institute, Wildau, Germany

[4]Genome Sequencing and Genomic Epidemiology, Robert Koch Institute, Berlin, Germany

[5]Advanced Light and Electron Microscopy, Robert Koch Institute, Berlin, Germany

[6]National Reference Laboratory for Campylobacter, Department of Biological Safety, German Federal Institute for Risk Assessment (BfR), Berlin, Germany

## AUTHOR ORCIDs

Jens-Uwe Ulrich http://orcid.org/0000-0002-4349-4843
Torsten Semmler http://orcid.org/0000-0002-2225-7267
Bernhard Y. Renard http://orcid.org/0000-0003-4589-9809

## FUNDING

| Funder | Grant(s) | Author(s) |
| --- | --- | --- |
| Bundesministerium für Bildung und Forschung (BMBF) | 031L0175B, 01KI1905D | Jens-Uwe Ulrich |
| Deutsches Zentrum für Infektionsforschung (DZIF) | TI 06.904 - FP2019 | Jens-Uwe Ulrich |
| Deutsche Forschungsgemeinschaft (DFG) | 425959793 | Lennard Epping |
| Deutsche Forschungsgemeinschaft (DFG, German Research Foundation) | 491466077 | Jens-Uwe Ulrich |

## AUTHOR CONTRIBUTIONS

Jens-Uwe Ulrich, Conceptualization, Data curation, Formal analysis, Methodology, Software, Visualization, Writing – original draft | Lennard Epping, Conceptualization, Resources, Validation, Writing – review and editing | Tanja Pilz, Methodology, Resources | Birgit Walther, Methodology, Resources | Kerstin Stingl, Methodology, Resources | Torsten Semmler, Conceptualization, Funding acquisition, Resources, Supervision, Writing – review and editing | Bernhard Y. Renard, Conceptualization, Funding acquisition, Investigation, Project administration, Resources, Supervision, Writing – review and editing

## DATA AVAILABILITY

The scripts and software used in the analysis are available in the GitHub repository, as described in Materials and Methods. The sequence data sets generated and analyzed during the current study are available in the NCBI BioProject repository under accession number PRJNA862336.

## ADDITIONAL FILES

The following material is available online.

### Supplemental Material

**Supplemental Material (mSystems00945-23-s0001.pdf).** Figures S1-S9 and Table S1.
**Table S2 (mSystems00945-23-s0002.xlsx).** Overview of sequenced bacterial isolates.
**Table S3 (mSystems00945-23-s0003.xlsx).** Overview of sequenced bacterial isolates.

### Open Peer Review

**PEER REVIEW HISTORY (review-history.pdf).** An accounting of the reviewer comments and feedback.

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
