## [Reviewer comments · mSystems]

Nanopore adaptive sampling effectively enriches bacterial plasmids

Jens-Uwe Ulrich, Lennard Epping, Tanja Pilz, Birgit Walther, Kerstin Stingl, Torsten Semmler, and Bernhard Renard

Corresponding Author(s): Jens-Uwe Ulrich, Hasso-Plattner-Institut für Digital Engineering gGmbH

Review Timeline:

Submission Date:	September 4, 2023
Editorial Decision:	November 10, 2023
Revision Received:	December 14, 2023
Accepted:	January 23, 2024

Editor: Juliette Hayer

Reviewer(s): The reviewers have opted to remain anonymous.

Transaction Report:

DOI: <https://doi.org/10.1128/msystems.00945-23>

Re: mSystems00945-23 (Nanopore adaptive sampling effectively enriches bacterial plasmids)

Dear Mr. Jens-Uwe Ulrich:

Please take into account the reviewers valuable comments, I think the manuscript will greatly benefit from addressing the questions they raised.

I would notably recommend the authors to clarify the information regarding the expired flow cells, and also to consider adding an analysis to assess more clearly the potential performance differences between MinKNOW and ReadBounce, as suggested by both reviewers.

Revision Guidelines

Sincerely,
Juliette Hayer
Editor
mSystems

Reviewer #1 (Comments for the Author):

Ulrich et al. have submitted a manuscript describing an evaluation of the Oxford Nanopore adaptive read sampling technique applied to plasmid enrichment in bacterial samples. Accurate plasmid assembly is important for a number of reasons, including, as noted by the authors, the role played by plasmids in the spread of antibiotic resistance elements. The question of whether plasmid assembly can be improved by adaptive sampling is therefore a relevant one, likely of interest to a large community, and the manuscript meaningfully contributes to the current state of knowledge in regard to this question. Prior to publication, however, a number of issues should be addressed, mostly concerning i) a streamlining of the "Results" section and a clearer presentation of the most important metrics, ii) making important experimental details and the rationale of the presented study more clear, and iii) the addition of a small number of additional analyses.

Major:

- Some important experimental details should be described more clearly. This includes, first and foremost, the genomes of the bacterial isolates used for the comparison - how many plasmids are present in the sequenced bacteria and what are their properties (e.g. size, approximate copy number, GC content etc.)? Second, the authors devote substantial part of their manuscript to describing the performance of "expired" flow cells in general and in the context of adaptive sampling, without giving much details about the expiration status of the utilised cells - which assumedly just means that the sequencing runs were carried out after the flow cell expiration date specified by Oxford Nanopore? If so, by how much? Third, the study is unclear about which references were actually used for the read depletion process - the Methods section says that the "chromosomal references of the bacterial isolates" were used as depletion targets, which suggests the availability of de novo assemblies for the specific isolates used in this study? Or do the authors mean that they used reference genomes from e.g. GenBank or RefSeq (and if so, how did the authors select which of the many available genomes for some of the species to use)?
- The rationale of the study should be described more clearly (also relating to the point about which references were used in the study). If strain-specific de novo assemblies were used for depletion, in which situation would a researcher typically expect to be in possession of a high-quality (long-read-sequencing based?) de novo assembly that faithfully represents the chromosomal, but not the plasmid, components of the bacterial isolates of interest? The authors write that "these [long-read sequencing] methods suffer from the small proportion of plasmid DNA within the sequenced samples... primarily if bacteria can not be cultivated in the lab" - fair enough, however i) the *coverage* of plasmids is often higher than that of the chromosomal genome, because of the oft-increased >1 copy number of counts of plasmids, and *coverage* is a much more relevant metric for de novo assembly than absolute proportion, and ii) the (interesting) case of non-culturable bacteria is not addressed in the presented study at all. Conversely, if a strain-specific reference genome is not available, wouldn't one want to characterize both chromosomal and plasmid genomes in most situations? I.e. when would one accept a (relatively modest) improvement in absolute plasmid sequencing yield at the cost of a substantially reduced yield on the chromosomal genome? In addition, if the general setting of the study is to use generic reference genomes, it would be important to quantify and discuss the impact of mismatches between the generic reference genome and the sequenced strain.
- The most important factor (at read length held constant) influencing assembly is absolute coverage, but the paper does not have a good figure comparing achieved plasmid coverages between the "adaptive sampling" / "no adaptive sampling" scenarios. Figure 2b shows the relevant data, but because the Y axis in the lower panel is dominated by *C. jejuni*, it is hard to assess the increase in absolute coverage achieved by adaptive sampling for the other species - for example, for *C. coli* in MinKNOW1, is there much of an improvement at all? Figure 6 shows relative enrichment (which is interesting, but not as relevant as the difference in absolute coverage); Figure 8 is informative about the extent of relative absolute enrichment (which is relevant), but is not informative about absolute coverage. We would recommend adding - as this is really the most important message of the paper - a main figure that contains the same data as existing Figure 2b, but with variable Y axes.
- The Results section is much too verbose. Figure 8, the first figure that is effectively informative about the achieved degree of absolute enrichment, is placed at the very end of the Results section. We would recommend re-structuring the section with a clear focus on and drive towards the most important questions of absolute plasmid coverage and plasmid assembly quality. Commenting on e.g. the read length effect of adaptive sampling is important, but this could be handled in one or two sentences with reference to a supplementary figure. The section "Reduced sequencing yield but same data quality for expired flow cells" is similarly way too long (also because the question of the effect of flow cell "expiration" is, in the context of this paper, only relevant to the extent that it influences the performance of adaptive sampling - for any generalizable conclusions about the effect of flow cell expiration, an $n = 1 / 3$ is way too small).
- The plasmid assembly quality comparison should be extended to include all species. The authors write "we did not include *K. pneumoniae* because of the findings mentioned in the last subsection that could bias analysis" - homologies between the chromosomal and plasmid genomes may indeed *influence* the analysis, but, as such homologies are a systematic factor that will often be present in real bacterial isolates, bias is created by ignoring the affected isolates, but by including them. The other two species were not included because "assembly statistics would not be comparable between the sequencing runs ReadBouncer1 and MinKNOW1" - that only applies to differences in absolute coverage, but not to the relative effect of using adaptive sampling on plasmid assembly coverage.
- It is very hard to interpret the presented data with respect to potential performance differences between MinKNOW and ReadBouncer - being able to make recommendations on which approach to use would certainly be very useful. Could an

analysis that answers that question be added?

- In the "Result" section it is stated that mean Phred scores were equal among all experiments with ReadBouncer1 and MinKNOW2 having significant fractions of reads with scores between 5 and 7. Looking at Fig. 5 it seems that both MinKNOW runs have larger fractions of low quality reads - this section should be revisited.

Minor:

- It would be good to include details on the identified homologous region between chromosomal genome and plasmid
- Some of the definitions in the paper are a bit hard to follow. "We refer to the percentage of plasmid base pairs as the relative plasmid abundance in a sample", "We calculated the enrichment by composition by dividing the relative plasmid abundance from adaptive sampling regions by the relative plasmid abundance from control regions", "We calculated the enrichment by yield using the number of sequenced base pairs from adaptive sampling and control regions" - it is not immediately clear what the difference between these metrics is. Perhaps labelling them as "Relative enrichment" and "absolute enrichment" may be more intuitive? Incidentally, what is the model by Martin et al. supposed to predict - enrichment by yield or compositional enrichment?
- With the aforementioned definitions some complex word combinations appear. Such as: "we do not see higher relative yield reduction" (line 173), "produces less sequencing yield" (means yielding less) (line 233)
- It would be good comment on why different DNA extraction kits were used for species belonging to the same genus
- The wording "...we separately sequenced five bacterial strains..." is a bit misleading as there were only 2 sequencing runs
- The authors suggest that sample handling, preparation and chosen barcode have a greater effect on read length than using expired flow cells - this statement should be removed or more explicitly tied to the the presented data
- The authors claim that adaptive sampling "enables plasmid assembly even after 2 hours of sequencing"; while Table 2 shows that adaptive sampling improved plasmid assembly quality, it also shows that assembly was possible without adaptive sampling. The word "enables" should thus be removed.
- In Figure 6, chromosome and plasmid abundances do not always add up to 100%. What is the reason for this? It would be worth explaining this in the text. Also, for consistency with the definition in lines 191-192, the title of the figure should be "plasmid relative abundances" and not "plasmid abundances"

Reviewer #2 (Comments for the Author):

In this manuscript Ulrich et al. describe an application of the relatively novel technique of targeted sequencing by purely computational means, called adaptive sampling. This method allows users to specify genomic coordinates from which reads are either to be sequenced or rejected from nanopores after reading some initial amount of data and comparing their origin to the specified coordinates. Rejection of a DNA molecule then allows for a different one to be captured, which can save time and therefore lead to either enrichment of certain sequences or depletion of others.

In this article the authors demonstrate how this technology can be used to increase the yield from bacterial plasmids by depleting all sequences that stem from the bacterial chromosome. They show that this can be achieved with two different implementations of adaptive sampling (one of them previously published by the authors) and the other available in Oxford Nanopore Technologies MinKNOW software. Furthermore, the results demonstrate that expired flowcells can be deployed for experiments using adaptive sampling without significant detriment compared to standard sequencing.

The article is overall well-written and all analyses appear sound with conclusions justified by the presented data. References to previous work are appropriate and up-to-date. Code and data is available as stated in the manuscript. It is a valuable work not only to show that adaptive sampling can be useful for sequencing plasmids, but also to give approximate numbers of what enrichment/depletion can be expected when using this technique with different bacteria and plasmid abundances. I especially appreciate the comparison to the theoretical work of Martin et al. (2020). I do not see any major issues with this work, but have several comments for the authors to consider:

General comments:

In several places (e.g. lines 126, 172) the authors explain decreased yield when using adaptive sampling by an additional time that is spent waiting to capture novel fragments at pores compared to the situation of sequencing without adaptive sampling. It makes sense that some time is needed to effect rejections, i.e. to reverse the voltage polarity to eject a molecule from a pore. However, it is not clear why an additional time penalty for capturing new molecules should be incurred as well. Indeed, if the two halves of the flowcell (acting as control and using adaptive sampling) are saturated in terms of DNA sample concentrations, then it would be expected that new DNA strands are captured at the same rate. Could the authors please clarify this point?

Regarding the degradation of actively sequencing channels, the authors describe a consistently lower number of channels on the adaptively sampling section of the flowcell. In the discussion (line 310), they conclude that this reflects faster degradation of flowcells when using adaptive sampling. There is undoubtedly a difference between the two sampling schemes seen in Fig. 4a. However, I am not convinced that this necessarily indicates faster degradation. In fact, in some cases it seems that the adaptive sampling section had fewer active pores at the start of the experiment, i.e. the intercept on the y-axis at timepoint 0. I would therefore suggest some additional analysis of the slope of this data, since the slope corresponds to the rate of degradation and could account for the differential numbers of pores at the beginning. Additionally, the statement of "1.4 -2.6x" active pores (line 180) could benefit from some clarification of what timepoints and experiments this corresponds to and whether any difference from the starting conditions was accounted for.

Assemblies of plasmids were generated by metaFlye, intended for metagenomic assemblies. Given that the sequencing was performed on bacterial isolates/data was demultiplexed by species, it is unclear why the "meta" mode of Flye was chosen. Could the authors please clarify their motivation or indicate whether any differences were observed compared to using Flye without the "--meta" flag?

It is not clear what the comparison of ReadBouncer and MinKNOW achieves in the manuscript. In the discussion the authors mention that it is not intended as a benchmark, which might leave the reader wondering what the purpose of the comparison is. While it is interesting to see two independent implementations of the technique achieve similar results, it would also be interesting to have some additional discussion of the differences and in which scenarios one or the other might achieve better results; or which parameters could be tweaked with either method to adapt it to sample-specific variables and potentially enhance the benefit of using either one. This could greatly help readers decide which of the implementations to use in their experiments.

Throughout the manuscript the authors describe the use of "expired" flowcells. It would be very helpful to specify what that means (manufacturers maximum recommended storage time?) and to quantify how far past their "expiration date" the used flowcells were, as "expiration" could indicate any arbitrary timeframe. Further, on line 305 of the Discussion, the authors mention the re-use of a flowcell. Were the flowcells expired and/or re-used, and does this apply to all or only one of them? How does that influence the results?

I don't suggest the authors need to perform this experiment, but would it be technically possible to run adaptive sampling with ReadBouncer and MinKNOW, plus a control section without adaptive sampling on a single flowcell? This could eliminate much of the issues discussed in the manuscript, such as differential pore availability, differences in sample concentration, etc.

Specific comments:

Line 69: The authors describe adaptive sampling as an in-silico technique. The methodology is indeed purely computational and does not require any laboratory preparation, yet the term is usually used for computer simulations, i.e. experiments conducted entirely on computer silicon. I am therefore wondering if the usage is intentional in this case?

It would be useful to briefly describe the plasmids contained in the different samples near the start of the results section, i.e. how many plasmids are present and what their expected sequence length is. This would help readers interpret read lengths, enrichment, and assembly statistics throughout the manuscript.

Line 113: The first reference to a figure in the text is to Fig. 1c, and should probably reference Fig. 1b instead according to the text. Additionally, it is slightly odd to not reference panel a first. Indeed, I can't seem to find a reference to that panel in the text.

Fig. 1: Following the comment above, I suggest reordering panels to correspond better to the order of appearance in the text. Additionally, I suggest the addition of an explanation of the elements of the boxplots shown in panel a (or mention that it's the standards of ggplot). For current panel c it would also be interesting to see the densities of read lengths for the adaptive sampling-section of the flowcell to see the variance of the length of rejected reads.

Lines 117-122: The statements given in this section could benefit from referencing the figure panels to guide the reader. Further, it would be helpful to state the duration of the sequencing run near the beginning of the results section to associate the yield values presented with the amount of time of data generation.

Fig. 2 caption: I suggest using "separated" instead of "divided" to explain the data shown in this figure. Readers might expect to see ratios of enrichment when mentioning "division", which are in fact presented later on in the article.

Line 132: The authors state that more reads were rejected by the "ReadBouncer2" flowcell compared to "ReadBouncer1". This is not intuitive considering the large difference in available pores between the two flowcells and a (presumably) equal duration of the experiment. This statement could be aided by some additional explanation.

Line 156-157: The authors suggest that the used barcodes, among other reasons, might influence the obtained read lengths. This is not clear to me (as a computational biologist) and might benefit from additional clarification.

Fig. 5: Both panels of results for ReadBouncer show two distinct densities for read lengths of the rejected DNA fragments. I could not find any discussion of this observation in the text or caption and am wondering if the authors have an explanation or hypothesis about this?

Fig. 5: The x-axis labels indicate \log_{10} , when the presented values seem to not be log-values but actual read lengths.

Lines 205-206: Here, the authors argue that MinKNOW achieves higher levels of enrichment due to faster rejection decisions. It could be helpful to be more precise in this statement, i.e. is the higher enrichment due to using less sequence data for alignment, is it due to differences in speed of performing the alignment, or perhaps due to other factors influencing the processing of read rejections?

Fig. 7: Panels b and d are comparisons of observed and predicted values. To ease the interpretation of the data, it would help to either make sure the panels have a square aspect ratio, to add a diagonal line to the plot, or both. In panels a and c, a horizontal line could be added to indicate the ratio of 1. Changing the y-axis range in panel c would also help the reader in discerning differences and interpreting results. The caption text of panel c contains the same description as panel a. This redundant text could be replaced with more informative text that aids in understanding the results.

Line 208ff: The authors state that the original plasmid abundance does not impact the enrichment by yield, only the enrichment by composition. Could an explanation of this be added? Is this due to the difference of read lengths from chromosomal material and plasmid sequencing reads? It would in general be interesting to see the read length distribution separated by plasmid and chromosomal DNA; or whether there is any difference given the plasmid sizes (not mentioned in the manuscript, see above).

Lines 216ff: When referring to equations it would be helpful to reference the methods section in which they appear, especially since the methods section is printed after the results in this version of the manuscript.

Lines 219, 223: Panel b of Fig. 8 is referenced before panel a. It might be worth swapping the figure panels if that is more natural to how the results are presented in the text.

Fig. 8: As in the previous figure, it might be helpful to add a line at the ratio of 1. Additionally, it would be interesting to have an interpretation of the shape of the results shown here, i.e. the steep increase in the beginning, followed by a steady decline of the advantage as the experiment progresses. Is this related to a possibly increased pore degradation, depletion of plasmid sequence material, or any other factors?

Line 271: The authors state that adaptive sampling improved plasmid assemblies "in all cases". The results for *Salmonella enterica* of the left MinKNOW column, however, indicate that the control assembly of the plasmid is in fact better than the adaptive sampling assembly.

Line 402: I assume the references used for adaptive sampling correspond to the accessions indicated in the section "Culture and DNA extraction"? It would be helpful to clarify.

Line 417: Raw data collected in 0.4s were used for real-time basecalling for both methods. Given the sequencing speed of 420 nucleotides/s stated in the manuscript, this corresponds to ~168nt. On lines 343-344 the authors discuss that unique sequences to discern similar segments would need to be contained in the first 1000nt. If only ~168nt are used to make decisions, would unique sequences not have to be present in that amount of data?

In this manuscript Ulrich et al. describe an application of the relatively novel technique of targeted sequencing by purely computational means, called adaptive sampling. This method allows users to specify genomic coordinates from which reads are either to be sequenced or rejected from nanopores after reading some initial amount of data and comparing their origin to the specified coordinates. Rejection of a DNA molecule then allows for a different one to be captured, which can save time and therefore lead to either enrichment of certain sequences or depletion of others.

In this article the authors demonstrate how this technology can be used to increase the yield from bacterial plasmids by depleting all sequences that stem from the bacterial chromosome. They show that this can be achieved with two different implementations of adaptive sampling (one of them previously published by the authors) and the other available in Oxford Nanopore Technologies MinKNOW software. Furthermore, the results demonstrate that expired flowcells can be deployed for experiments using adaptive sampling without significant detriment compared to standard sequencing.

The article is overall well-written and all analyses appear sound with conclusions justified by the presented data. References to previous work are appropriate and up-to-date. Code and data is available as stated in the manuscript. It is a valuable work not only to show that adaptive sampling can be useful for sequencing plasmids, but also to give approximate numbers of what enrichment/depletion can be expected when using this technique with different bacteria and plasmid abundances. I especially appreciate the comparison to the theoretical work of Martin et al. (2020). I do not see any major issues with this work, but have several comments for the authors to consider:

General comments:

- In several places (e.g. lines 126, 172) the authors explain decreased yield when using adaptive sampling by an additional time that is spent waiting to capture novel fragments at pores compared to the situation of sequencing without adaptive sampling. It makes sense that some time is needed to effect rejections, i.e. to reverse the voltage polarity to eject a molecule from a pore. However, it is not clear why an additional time penalty for capturing new molecules should be incurred as well. Indeed, if the two halves of the flowcell (acting as control and using adaptive sampling) are saturated in terms of DNA sample concentrations, then it would be expected that new DNA strands are captured at the same rate. Could the authors please clarify this point?
- Regarding the degradation of actively sequencing channels, the authors describe a consistently lower number of channels on the adaptively sampling section of the flowcell. In the discussion (line 310), they conclude that this reflects faster degradation of flowcells when using adaptive sampling. There is undoubtedly a difference between the two sampling schemes seen in Fig. 4a. However, I am not convinced that this necessarily indicates faster degradation. In fact, in some cases it seems that the adaptive sampling section had fewer active pores at the start of the experiment, i.e. the intercept on the y-axis at timepoint 0. I would therefore suggest some additional analysis of the slope of this data, since the slope corresponds to the rate of degradation and could account for the differential numbers of pores at the beginning. Additionally, the statement of “1.4 -2.6x” active pores (line 180) could

benefit from some clarification of what timepoints and experiments this corresponds to and whether any difference from the starting conditions was accounted for.

- Assemblies of plasmids were generated by metaFlye, intended for metagenomic assemblies. Given that the sequencing was performed on bacterial isolates/data was demultiplexed by species, it is unclear why the “meta” mode of Flye was chosen. Could the authors please clarify their motivation or indicate whether any differences were observed compared to using Flye without the “--meta” flag?
- It is not clear what the comparison of ReadBouncer and MinKNOW achieves in the manuscript. In the discussion the authors mention that it is not intended as a benchmark, which might leave the reader wondering what the purpose of the comparison is. While it is interesting to see two independent implementations of the technique achieve similar results, it would also be interesting to have some additional discussion of the differences and in which scenarios one or the other might achieve better results; or which parameters could be tweaked with either method to adapt it to sample-specific variables and potentially enhance the benefit of using either one. This could greatly help readers decide which of the implementations to use in their experiments.
- Throughout the manuscript the authors describe the use of “expired” flowcells. It would be very helpful to specify what that means (manufacturers maximum recommended storage time?) and to quantify how far past their “expiration date” the used flowcells were, as “expiration” could indicate any arbitrary timeframe. Further, on line 305 of the Discussion, the authors mention the re-use of a flowcell. Were the flowcells expired and/or re-used, and does this apply to all or only one of them? How does that influence the results?
- I don't suggest the authors need to perform this experiment, but would it be technically possible to run adaptive sampling with ReadBouncer and MinKNOW, plus a control section without adaptive sampling on a single flowcell? This could eliminate much of the issues discussed in the manuscript, such as differential pore availability, differences in sample concentration, etc.

Specific comments:

- Line 69: The authors describe adaptive sampling as an *in-silico* technique. The methodology is indeed purely computational and does not require any laboratory preparation, yet the term is usually used for computer simulations, i.e. experiments conducted entirely on computer *silicon*. I am therefore wondering if the usage is intentional in this case?
- It would be useful to briefly describe the plasmids contained in the different samples near the start of the results section, i.e. how many plasmids are present and what their expected sequence length is. This would help readers interpret read lengths, enrichment, and assembly statistics throughout the manuscript.

- Line 113: The first reference to a figure in the text is to Fig. 1c, and should probably reference Fig. 1b instead according to the text. Additionally, it is slightly odd to not reference panel a first. Indeed, I can't seem to find a reference to that panel in the text.
- Fig. 1: Following the comment above, I suggest reordering panels to correspond better to the order of appearance in the text. Additionally, I suggest the addition of an explanation of the elements of the boxplots shown in panel a (or mention that it's the standards of ggplot). For current panel c it would also be interesting to see the densities of read lengths for the adaptive sampling-section of the flowcell to see the variance of the length of rejected reads.
- Lines 117-122: The statements given in this section could benefit from referencing the figure panels to guide the reader. Further, it would be helpful to state the duration of the sequencing run near the beginning of the results section to associate the yield values presented with the amount of time of data generation.
- Fig. 2 caption: I suggest using "separated" instead of "divided" to explain the data shown in this figure. Readers might expect to see ratios of enrichment when mentioning "division", which are in fact presented later on in the article.
- Line 132: The authors state that more reads were rejected by the "ReadBouncer2" flowcell compared to "ReadBouncer1". This is not intuitive considering the large difference in available pores between the two flowcells and a (presumably) equal duration of the experiment. This statement could be aided by some additional explanation.
- Line 156-157: The authors suggest that the used barcodes, among other reasons, might influence the obtained read lengths. This is not clear to me (as a computational biologist) and might benefit from additional clarification.
- Fig. 5: Both panels of results for ReadBouncer show two distinct densities for read lengths of the rejected DNA fragments. I could not find any discussion of this observation in the text or caption and am wondering if the authors have an explanation or hypothesis about this?
- Fig. 5: The x-axis labels indicate \log_{10} , when the presented values seem to not be log-values but actual read lengths.
- Lines 205-206: Here, the authors argue that MinKNOW achieves higher levels of enrichment due to faster rejection decisions. It could be helpful to be more precise in this statement, i.e. is the higher enrichment due to using less sequence data for alignment, is it due to differences in speed of performing the alignment, or perhaps due to other factors influencing the processing of read rejections?
- Fig. 7: Panels b and d are comparisons of observed and predicted values. To ease the interpretation of the data, it would help to either make sure the panels have a square aspect ratio, to add a diagonal line to the plot, or both. In panels a and c, a

horizontal line could be added to indicate the ratio of 1. Changing the y-axis range in panel c would also help the reader in discerning differences and interpreting results. The caption text of panel c contains the same description as panel a. This redundant text could be replaced with more informative text that aids in understanding the results.

- Line 208ff: The authors state that the original plasmid abundance does not impact the enrichment by yield, only the enrichment by composition. Could an explanation of this be added? Is this due to the difference of read lengths from chromosomal material and plasmid sequencing reads? It would in general be interesting to see the read length distribution separated by plasmid and chromosomal DNA; or whether there is any difference given the plasmid sizes (not mentioned in the manuscript, see above).
- Lines 216ff: When referring to equations it would be helpful to reference the methods section in which they appear, especially since the methods section is printed after the results in this version of the manuscript.
- Lines 219, 223: Panel b of Fig. 8 is referenced before panel a. It might be worth swapping the figure panels if that is more natural to how the results are presented in the text.
- Fig. 8: As in the previous figure, it might be helpful to add a line at the ratio of 1. Additionally, it would be interesting to have an interpretation of the shape of the results shown here, i.e. the steep increase in the beginning, followed by a steady decline of the advantage as the experiment progresses. Is this related to a possibly increased pore degradation, depletion of plasmid sequence material, or any other factors?
- Line 271: The authors state that adaptive sampling improved plasmid assemblies “in all cases”. The results for *Salmonella enterica* of the left MinKNOW column, however, indicate that the control assembly of the plasmid is in fact better than the adaptive sampling assembly.
- Line 402: I assume the references used for adaptive sampling correspond to the accessions indicated in the section “Culture and DNA extraction”? It would be helpful to clarify.
- Line 417: Raw data collected in 0.4s were used for real-time basecalling for both methods. Given the sequencing speed of 420 nucleotides/s stated in the manuscript, this corresponds to ~168nt. On lines 343-344 the authors discuss that unique sequences to discern similar segments would need to be contained in the first 1000nt. If only ~168nt are used to make decisions, would unique sequences not have to be present in that amount of data?

Dear Dr. Juliette Hayer,

We appreciate the thoughtful comments by you and the reviewers, which help us to improve our work on the enrichment of bacterial plasmids via nanopore adaptive sampling.

We have adjusted our manuscript according to the reviewers' comments. In particular, we shortened the first sections of the results part and moved the descriptions and figures to the supplements as recommended by reviewer #1. Further, we added more details on the sequenced bacterial plasmids and information regarding the expired flow cells. We also extended the de novo plasmid assemblies table to all bacterial samples, showing all assemblies for the first 3 hours of sequencing. Since this table is too large for inclusion in the main manuscript, we added it as a supplemental file. In this context, we also commented on using the metagenomic assembler for plasmid assemblies in the manuscript and the point-by-point response below.

We modified and added several figures in the manuscript as recommended by both reviewers, including the zoom-in version of the plasmid yield increase by adaptive sampling (Figure 2) or separate read-length histograms for chromosomal and plasmid reads for each sample (Figures S5-S8). We also added a figure that shows the length differences of rejected reads separated by adaptive sampling tool to demonstrate the potential performance differences between MinKNOW and ReadBouncer.

Finally, we also extended the data analysis in the manuscript. In particular, we performed a Mann-Whitney U-Test to determine the statistically significant difference between average read quality scores and an investigation of the homologous regions that are located on both the *K. pneumoniae* chromosome and plasmid sequences.

Below, we give a point-by-point response to further issues raised by the reviewers.

Please do not hesitate to let us know if you require any further information from our side.

Point-by-point response

Reviewer #1:

Major:

- Some important experimental details should be described more clearly. This includes, first and foremost, the genomes of the bacterial isolates used for the comparison - how many plasmids are present in the sequenced bacteria and what are their properties (e.g. size, approximate copy number, GC content etc.)? Second, the authors devote substantial part of their manuscript to describing the performance of "expired" flow cells in general and in the context of adaptive sampling, without giving much details about the expiration status of the utilised cells - which assumedly just means that the sequencing runs were carried out after the flow cell expiration date specified by Oxford Nanopore? If so, by how much? Third, the study is unclear about which references were actually used for the read depletion process - the Methods section says that the "chromosomal references of the bacterial isolates" were used as depletion targets, which suggests the availability of de novo assemblies for the specific isolates used in this study? Or do the authors mean that they used reference genomes from e.g. GenBank or RefSeq (and if so, how did the authors select which of the many available genomes for some of the species to use)?

As suggested by the reviewer, we added supplemental table S1, which summarizes the bacterial isolates sequenced in this study. We added NCBI RefSeq IDs, sizes of chromosomes and plasmids as well as the approximate plasmid copy number. Secondly, we added the following statement to the manuscript, clarifying the flow cells' expiration status: "All flow cells, except ReadBouncer1, were used 2-3 months after reaching the manufacturer's recommended storage duration, and throughout the manuscript, we will refer to the flow cells according to the adaptive sampling tool used."

Regarding the third statement, we added NCBI Ref IDs in the methods section and to the supplemental table S1. Some of the references were sequenced, assembled and published by authors of the manuscript. All bacteria were selected because of their clinical relevance and known antimicrobial resistance gene harboring plasmids, making them reasonable study objects for the enrichment of ARG-harboring plasmids via adaptive sampling. Thus, we added the following statement to the results section:

"All of the chosen bacterial strains are clinically important human pathogens with ARG harboring plasmids, which have already been sequenced in our laboratory. Thus, their chromosomal and plasmid reference sequences were sequenced, assembled, and characterized before conducting this study, which provided us with the necessary ground truth for our data analysis."

- The rationale of the study should be described more clearly (also relating to the point about which references were used in the study). If strain-specific de novo assemblies were used for depletion, in which situation would a researcher typically expect to be in possession of a high-quality (long-read-sequencing based?) de novo assembly that faithfully represents the chromosomal, but not the plasmid, components of the bacterial isolates of interest? The

authors write that "these [long-read sequencing] methods suffer from the small proportion of plasmid DNA within the sequenced samples... primarily if bacteria can not be cultivated in the lab" - fair enough, however i) the *coverage* of plasmids is often higher than that of the chromosomal genome, because of the oft-increased >1 copy number of counts of plasmids, and *coverage* is a much more relevant metric for de novo assembly than absolute proportion, and ii) the (interesting) case of non-culturable bacteria is not addressed in the presented study at all. Conversely, if a strain-specific reference genome is not available, wouldn't one want to characterize both chromosomal and plasmid genomes in most situations? I.e. when would one accept a (relatively modest) improvement in absolute plasmid sequencing yield at the cost of a substantially reduced yield on the chromosomal genome? In addition, if the general setting of the study is to use generic reference genomes, it would be important to quantify and discuss the impact of mismatches between the generic reference genome and the sequenced strain.

We agree with the reviewer that the statement about bacteria that cannot be cultivated in the lab may be misleading and thus removed it from the manuscript. The study's rationale is more about investigating a new technology for the enrichment of plasmids from well-known, clinically relevant human pathogens. In a hospital setting with an increasing risk of nosocomial infections where we know the pathogen, we are primarily interested in antibiotic-resistance genes, which are mainly located on plasmids. In such cases, a database of high-quality reference genomes of bacterial pathogens could be used as a depletion target to investigate whether there is a risk of AMR harboring plasmids that could impact the treatment of patients. We regard this study as a proof-of-concept to examine whether plasmid enrichment is possible in the most straightforward use case. However, de novo sequencing of bacterial isolates would not be the intended use case. Thus, we added the second sentence to the following statement, which should clarify the rationale of the study:

"Thus, additional sample preparation steps are required to isolate or enrich plasmids before DNA sequencing, but they are too expensive and laborious for applications in clinical diagnostic settings. These are particularly interesting for nosocomial infections where the potential pathogens are known, and the focus lies on identifying antibiotic resistance genes, which are mainly present on plasmids and could impact the treatment of patients."

- The most important factor (at read length held constant) influencing assembly is absolute coverage, but the paper does not have a good figure comparing achieved plasmid coverages between the "adaptive sampling" / "no adaptive sampling" scenarios. Figure 2b shows the relevant data, but because the Y axis in the lower panel is dominated by *C. jejuni*, it is hard to assess the increase in absolute coverage achieved by adaptive sampling for the other species - for example, for *C. coli* in MinKNOW1, is there much of an improvement at all? Figure 6 shows relative enrichment (which is interesting, but not as relevant as the difference in absolute coverage); Figure 8 is informative about the extent of relative absolute enrichment (which is relevant), but is not informative about absolute coverage. We would recommend adding - as this is really the most important message of the paper - a main figure that contains the same data as existing Figure 2b, but with variable Y axes.

We agree with the reviewer that the increase in absolute plasmid yield is hard to assess from the figure. We followed the reviewer's recommendations and added the following figure to the manuscript

- The Results section is much too verbose. Figure 8, the first figure that is effectively informative about the achieved degree of absolute enrichment, is placed at the very end of the Results section. We would recommend re-structuring the section with a clear focus on and drive towards the most important questions of absolute plasmid coverage and plasmid assembly quality. Commenting on e.g. the read length effect of adaptive sampling is important, but this could be handled in one or two sentences with reference to a supplementary figure. The section "Reduced sequencing yield but same data quality for expired flow cells" is similarly way too long (also because the question of the effect of flow cell "expiration" is, in the context of this paper, only relevant to the extent that it influences the performance of adaptive sampling - for any generalizable conclusions about the effect of flow cell expiration, an $n = 1 / 3$ is way too small).

We appreciate the reviewer's comments on the length of the results section and moved a substantial part of the first two subsections into the Supplemental material. This includes the analysis of active pore counts, read lengths, and average read quality scores. We also moved the read length metrics table as well as all figures showing read length histograms, active pore counts and read quality scores to the supplements. This shortens the results section as recommended by the reviewer and shifts the focus to the major results of the study, which is the enrichment of plasmid sequences.

- The plasmid assembly quality comparison should be extended to include all species. The authors write "we did not include *K. pneumoniae* because of the findings mentioned in the last subsection that could bias analysis" - homologies between the chromosomal and plasmid genomes may indeed *influence* the analysis, but, as such homologies are a systematic factor that will often be present in real bacterial isolates, bias is created by ignoring the affected isolates, but by including them. The other two species were not included because "assembly statistics would not be comparable between the sequencing runs ReadBouncer1 and MinKNOW1" - that only applies to differences in absolute coverage, but not to the relative effect of using adaptive sampling on plasmid assembly coverage.

With regard to the reviewer's comment, we extended the assembly table to include de novo plasmid assemblies of all bacterial samples from all sequencing runs after one, two and three hours. Since the resulting table is too large for inclusion in the main manuscript, we decided to add the table as Supplemental Table S13. We further revisited the whole subsection that describes and interprets the results of the assembly comparison of adaptive sampling and control regions.

- It is very hard to interpret the presented data with respect to potential performance differences between MinKNOW and ReadBouncer - being able to make recommendations on which approach to use would certainly be very useful. Could an analysis that answers that question be added?

To better interpret the potential performance differences, we added a read length histogram of rejected reads, separated by adaptive sampling tool. Figure S12 is part of the Supplemental Material and shows that ReadBouncer generally uses longer read prefixes for the decision-making process, which results in later rejection decisions. However, in most cases, we don't see a significant difference in enrichment between the two methods, except for the depletion of *K. pneumoniae* in the MinKNOW experiment. Therefore, we added the following statement to the results section:

"An analysis of rejected reads revealed shorter read lengths for MinKNOW compared to ReadBouncer, which is caused by rejection decisions based on short read prefixes (see Figure S8). In the histogram, we see that reads rejected by ReadBouncer are longer than those rejected by MinKNOW, with an average length of 848 bp compared to 520 bp. This confirmed our assumption that ReadBouncer rejects reads later during the adaptive sampling process, resulting in a higher abundance of unwanted chromosomal base pairs in the final output. To avoid confusion, we have to note that the lengths of rejected reads in the final output are not the same as the read prefix (or chunk) length used by adaptive sampling tools for making rejection decisions. Lengths of rejected reads in the final output represent the time needed for the whole decision process, including time for communication with the API and mapping of reads against index data structures."

We further added the following statement to the discussion section:

"However, both tools consistently enriched low-abundant plasmid sequences, with only one exception where MinKNOW failed to enrich plasmid sequences for *Klebsiella pneumoniae*. Here, the fact that ReadBouncer uses longer read prefixes for the decision-making algorithm seems to prevent false rejection decisions. Unfortunately, the prefix length, alignment identity or minimum alignment length for decision-making cannot be parameterized via MinKNOW, which suggests that more tunable tools such as ReadBouncer are better suited for complex

samples. However, for less complex samples MinKNOW is more user-friendly and potentially achieves higher enrichment values by faster rejection decisions.”

- In the "Result" section it is stated that mean Phred scores were equal among all experiments with ReadBouncer1 and MinKNOW2 having significant fractions of reads with scores between 5 and 7. Looking at Fig. 5 it seems that both MinKNOW runs have larger fractions of low quality reads - this section should be revisited.

As recommended by the reviewer, we revisited the section before we moved the parts to the Supplemental Material. We agree that a significant fraction of reads has quality scores between 5 and 7 for the mentioned sequencing runs. Thus, we performed a Mann-Whitney U-Test to investigate whether there is a significant difference in average read quality scores between runs and between adaptive sampling and control regions of the same run. The analysis results are described in the Supplement, with the following statement:

“To further assess and compare the quality of the four sequencing runs, we look at the quality from the control regions of the sequencing runs. The contour plots in Figure S9 show that for all four sequencing runs, a large proportion of reads has a mean Phred quality between 12 and 15. Performing a Mann-Whitney U-Test revealed statistically significant differences in Phred quality scores between the different sequencing runs, but the effect sizes were small with regard to Cohen's classification ($r < 0.2$).

One of the major aspects of our study is the investigation of the impact adaptive sampling has on expired nanopore flow cells. Therefore, we first compared the average read quality scores from reads sequenced on control regions (Figure S9) with those sequenced on adaptive sampling regions (Figure S10). Although we see a statistically significant difference between adaptive sampling and control regions (Mann-Whitney U Test $p < 0.05$) for all 4 experiments, the effect sizes are very small with regard to Cohen's classification ($r < 0.1$).”

Minor:

- It would be good to include details on the identified homologous region between chromosomal genome and plasmid

We analyzed the identified regions and added the following part to the results section:

“An investigation of the annotated GenBank file revealed that two of those regions code for *IS6-like element IS26 family transposase* and *IS110-like element IS5075 family transposase*, both belonging to the group of insertion sequences (IS), which are small DNA segments (< 2kbp) that encode an enzyme, the transposase (Tnp), which catalyzes the DNA cleavage and strand-transfer reactions enabling the movement of the element between DNA molecules [25]. The third region codes for *group II intron reverse transcriptase/maturase*, a mobile genetic element encoding reverse transcriptases (RTs) that are important for RNA splicing (maturase activity) by helping the intron RNA fold into the catalytically active structure [26], and the fourth region encodes *CusA/CzcA family heavy metal efflux RND*, that is an efflux pump transporting heavy metal ions out of the bacterial cell and is important for antimicrobial resistance [27].”

- Some of the definitions in the paper are a bit hard to follow. "We refer to the percentage of plasmid base pairs as the relative plasmid abundance in a sample", "We calculated the enrichment by composition by dividing the relative plasmid abundance from adaptive sampling regions by the relative plasmid abundance from control regions", "We calculated the enrichment by yield using the number of sequenced base pairs from adaptive sampling and control regions" - it is not immediately clear what the difference between these metrics is. Perhaps labelling them as "Relative enrichment" and "absolute enrichment" may be more intuitive? Incidentally, what is the model by Martin et al. supposed to predict - enrichment by yield or compositional enrichment?

We thank the reviewer for the suggestions on more understandable definitions. We modified them and added the labels "relative" and "absolute" at places where it is unclear which of both was referred to. We also explicitly mention in the text that the mathematical model from Martin et al. is supposed to predict the relative compositional enrichment.

- With the aforementioned definitions some complex word combinations appear. Such as: "we do not see higher relative yield reduction" (line 173), "produces less sequencing yield" (means yielding less) (line 233)

We changed the wording as suggested by the reviewer.

- It would be good comment on why different DNA extraction kits were used for species belonging to the same genus

Here, the MagAttract HMW Genomic Extraction kit was used for internal validation in the sequencing facility we worked with. The statement has also been added to the Supplemental Material.

- The wording "...we separately sequenced five bacterial strains..." is a bit misleading as there were only 2 sequencing runs

We agree with the reviewer and changed the wording accordingly.

- The authors suggest that sample handling, preparation and chosen barcode have a greater effect on read length than using expired flow cells - this statement should be removed or more explicitly tied to the the presented data

Since we can not justify the statement with an in-depth analysis, we decided to remove the statement from the results section, as suggested by the reviewer.

- The authors claim that adaptive sampling "enables plasmid assembly even after 2 hours of sequencing"; while Table 2 shows that adaptive sampling improved plasmid assembly quality, it also shows that assembly was possible without adaptive sampling. The word "enables" should thus be removed.

We agree that the word "enables" is not appropriate and thus removed the statement.

- In Figure 6, chromosome and plasmid abundances do not always add up to 100%. What is the reason for this? It would be worth explaining this in the text. Also, for consistency with the definition in lines 191-192, the title of the figure should be "plasmid relative abundances" and not "plasmid abundances"

We agree that the percentages were not correctly calculated. We fixed the bug in the computation and updated the figure in the manuscript.

Reviewer #2 (Comments for the Author):

In several places (e.g. lines 126, 172) the authors explain decreased yield when using adaptive sampling by an additional time that is spent waiting to capture novel fragments at pores compared to the situation of sequencing without adaptive sampling. It makes sense that some time is needed to effect rejections, i.e. to reverse the voltage polarity to eject a molecule from a pore. However, it is not clear why an additional time penalty for capturing new molecules should be incurred as well. Indeed, if the two halves of the flowcell (acting as control and using adaptive sampling) are saturated in terms of DNA sample concentrations, then it would be expected that new DNA strands are captured at the same rate. Could the authors please clarify this point?

According to Oxford Nanopore Technologies, the time to capture a new DNA molecule is expected to be 0.5 seconds. Applying adaptive sampling results in the rejection of thousands to 100 thousands of molecules, leading to more molecules sequenced in the adaptive sampling region than in the control region. Thus, there is also more time spent capturing the DNA molecules on adaptive sampling regions during the same time interval of the

sequencing run. However, this will only partially explain the large overall yield differences we observe between adaptive sampling and control regions. For clarification, we added the following statement to the main manuscript:

“Assuming a read capturing time of 0.5 seconds and sequencing pace of 420 bp/second, the second point can account for up to 50 Mbp, if 250,000 additional reads are sequenced in the adaptive sampling region. However, this explains only a small fraction of the reduced yield, showing that fewer active sequencing channels are the main driver for the reduced overall yield. In general, we increased the yield in sequenced plasmid base pairs with adaptive sampling for all but one bacterial sample (Figure 1 (b) and Figure 2).”

Regarding the degradation of actively sequencing channels, the authors describe a consistently lower number of channels on the adaptively sampling section of the flowcell. In the discussion (line 310), they conclude that this reflects faster degradation of flowcells when using adaptive sampling. There is undoubtedly a difference between the two sampling schemes seen in Fig. 4a. However, I am not convinced that this necessarily indicates faster degradation. In fact, in some cases it seems that the adaptive sampling section had fewer active pores at the start of the experiment, i.e. the intercept on the y-axis at timepoint 0. I would therefore suggest some additional analysis of the slope of this data, since the slope corresponds to the rate of degradation and could account for the differential numbers of pores at the beginning. Additionally, the statement of "1.4 -2.6x" active pores (line 180) could benefit from some clarification of what timepoints and experiments this corresponds to and whether any difference from the starting conditions was accounted for.

We thank the reviewer for this feedback. Based on the suggestions, we performed a linear regression analysis of the data and added the resulting lines to the corresponding figure, which was moved to the Supplemental Material after shortening the results section as recommended by reviewer #1.

We further added the following statement to the “investigation of sequencing runs” section: “Although we consistently observe more active sequencing channels in control regions, our linear regression analysis did not reveal systematically faster degradation of pores in adaptive sampling regions. We could also not detect bigger systematic differences in active sequencing channels on expired flow cells when compared to the fresh flow cell ReadBouncer1.”

Assemblies of plasmids were generated by metaFlye, intended for metagenomic assemblies. Given that the sequencing was performed on bacterial isolates/data was demultiplexed by species, it is unclear why the "meta" mode of Flye was chosen. Could the authors please clarify their motivation or indicate whether any differences were observed compared to using Flye without the "--meta" flag?

There are two reasons why we chose to use a metagenomics assembler for the plasmid assembly. First, Johnson et al. (2023) have shown that ordinary long-read genome assemblers struggle with small plasmids. Second, having different plasmids with different copy numbers in a single cell is the same situation as we would expect in a metagenomics sample with different genomes at different copy numbers, resulting in uneven sequencing depths of contigs. The developers of Flye have mentioned this scenario on their GitHub page, and we added the following statement to the methods section for clarification: “Since most long-read assemblers struggle to correctly assemble small plasmids [38], we decided to use Flye/metaFlye assembler (v2.9.2, parameter "--meta") [38,29], which is

meant to improve the assembly of contigs with uneven sequence depths – a situation often experienced with plasmid sequences that are present at high copy numbers in a single cell.”

It is not clear what the comparison of ReadBouncer and MinKNOW achieves in the manuscript. In the discussion the authors mention that it is not intended as a benchmark, which might leave the reader wondering what the purpose of the comparison is. While it is interesting to see two independent implementations of the technique achieve similar results, it would also be interesting to have some additional discussion of the differences and in which scenarios one or the other might achieve better results; or which parameters could be tweaked with either method to adapt it to sample-specific variables and potentially enhance the benefit of using either one. This could greatly help readers decide which of the implementations to use in their experiments.

We chose to use a metagenomics assembler for the plasmid assembly for two reasons. First, Johnson et al. (2023) have shown that ordinary long-read genome assemblers struggle with small plasmids. Second, having different plasmids with different copy numbers in a single cell is the same situation as we would expect in a metagenomics sample with different genomes at different copy numbers, resulting in uneven sequencing depths of contigs. The developers of Flye have mentioned this scenario on their GitHub page, and we added the following statement to the methods section for clarification:

“Our study was by no means designed to benchmark different adaptive sampling tools, which would require the inclusion of more tools and a setup that ensures that all tools use the same amount of sequence information for making rejection decisions. This can only be ensured by using adaptive sampling simulation tools like Icarust or SimReadUntil. However, both tools consistently enriched low-abundant plasmid sequences, with only one exception where MinKNOW failed to enrich plasmid sequences for *Klebsiella pneumoniae*. Here, the fact that ReadBouncer uses longer read prefixes for the decision-making algorithm seems to prevent false rejection decisions. Unfortunately, the prefix length, alignment identity or minimum alignment length for decision-making cannot be parameterized via MinKNOW, which suggests that more tunable tools like ReadBouncer are better suited for complex samples. However, for less complex samples MinKNOW is more user-friendly and potentially achieves higher enrichment values through faster rejection decisions.”

Throughout the manuscript the authors describe the use of "expired" flowcells. It would be very helpful to specify what that means (manufacturers maximum recommended storage time?) and to quantify how far past their "expiration date" the used flowcells were, as "expiration" could indicate any arbitrary timeframe. Further, on line 305 of the Discussion, the authors mention the re-use of a flowcell. Were the flowcells expired and/or re-used, and does this apply to all or only one of them? How does that influence the results?

We appreciate the reviewer's comments and added a statement that clarifies the expiration status:

“All flow cells, except ReadBouncer1, were used 2-3 months after reaching the manufacturer's recommended storage duration, ...”

We further removed the “re-use” statement because none of the flow cells in the study were re-used. The only impact of the expiration we could observe was the lower number of active

sequencing pores throughout the experiments, which we also clarify in the following statement:

“Although the number of active sequencing pores on expired flow cells is generally below the minimum number of active pores covered by the manufacturer's warranty of 800 pores, we did not recognize a significant effect on read lengths and Phred quality scores.”

I don't suggest the authors need to perform this experiment, but would it be technically possible to run adaptive sampling with ReadBouncer and MinKNOW, plus a control section without adaptive sampling on a single flowcell? This could eliminate much of the issues discussed in the manuscript, such as differential pore availability, differences in sample concentration, etc.

This is a very interesting point by the reviewer, which we also considered. However, this is technically impossible because of the bi-directional connections between ONT's ReadUntil API and the used adaptive sampling tool. After establishing this connection, no second adaptive sampling tool can be connected to the same flow cell position. Even if the ReadUntil API would not refuse the connection, it would randomly send chunks of data to one of the two connected tools, which would lead to inconsistent data analysis.

Specific comments:

Line 69: The authors describe adaptive sampling as an in-silico technique. The methodology is indeed purely computational and does not require any laboratory preparation, yet the term is usually used for computer simulations, i.e. experiments conducted entirely on computer silicon. I am therefore wondering if the usage is intentional in this case?

The usage of the word “in-silico” was intended to reflect that the enrichment was solely achieved by a computational method without any wet-lab-based enrichment protocol. Therefore, we think that the wording makes sense in this context.

It would be useful to briefly describe the plasmids contained in the different samples near the start of the results section, i.e. how many plasmids are present and what their expected sequence length is. This would help readers interpret read lengths, enrichment, and assembly statistics throughout the manuscript.

As suggested by the reviewer, we added supplemental table S1, which summarizes the bacterial isolates sequenced in this study. We added NCBI RefSeq IDs, sizes of chromosomes and plasmids as well as the approximate plasmid copy number.

Line 113: The first reference to a figure in the text is to Fig. 1c, and should probably reference Fig. 1b instead according to the text. Additionally, it is slightly odd to not reference panel a first. Indeed, I can't seem to find a reference to that panel in the text.

We agree with the reviewer that this might need to be clarified for the reader and changed the figure panels and references in the text. Since reviewer #1 recommended shortening the first subsections of the results section, we moved the figure and the referencing text to the Supplemental Material.

Fig. 1: Following the comment above, I suggest reordering panels to correspond better to the order of appearance in the text. Additionally, I suggest the addition of an explanation of the elements of the boxplots shown in panel a (or mention that it's the standards of ggplot). For current panel c it would also be interesting to see the densities of read lengths for the adaptive sampling-section of the flowcell to see the variance of the length of rejected reads.

We thank the reviewer for the suggestions and reordered the panels in the figure. We further added the following explanation of the box plots to the figure caption:

“The line splitting the box represents the median read length. The lower edge of the box represents the lower quartile and the upper edge represents the upper quartile of the read length distributions.”

We further added read length density plots for each sample separated by plasmid and chromosome and by control and adaptive sampling region. Those figures were added to the Supplemental Material as well (Figures S5-S8).

Lines 117-122: The statements given in this section could benefit from referencing the figure panels to guide the reader. Further, it would be helpful to state the duration of the sequencing run near the beginning of the results section to associate the yield values presented with the amount of time of data generation.

As recommended by the reviewer, we are referencing the figure panel in the text and added a statement about the duration of the sequencing runs at the beginning of the results section:

“In this study, we present the application of nanopore adaptive sampling on the in-silico enrichment of plasmids by depleting chromosomal reads during the sequencing of bacterial isolates. Therefore, we sequenced five bacterial strains - *Campylobacter jejuni*, *Campylobacter coli*, *Salmonella enterica*, *Enterobacter hormaechei* and *Klebsiella pneumoniae* - on four different flow cells for 24 hours, each separated into an adaptive sampling and a control region.”

Fig. 2 caption: I suggest using "separated" instead of "divided" to explain the data shown in this figure. Readers might expect to see ratios of enrichment when mentioning "division", which are in fact presented later on in the article.

We followed the suggestion by the reviewer and changed the wording to “separated”.

Line 132: The authors state that more reads were rejected by the "ReadBouncer2" flowcell compared to "ReadBouncer1". This is not intuitive considering the large difference in available pores between the two flowcells and a (presumably) equal duration of the experiment. This statement could be aided by some additional explanation.

Thanks to the reviewer's comment, we noticed that the numbers of rejected reads, as stated in the text, were incorrect. Besides correcting the numbers, we also added a statement explaining the different numbers of reads between ReadBouncer1 and ReadBouncer2:

"In this context, we also see on all four flow cells a higher number of reads sequenced on the adaptive sampling regions than on the control regions (Figure 1 (c, d)). Thus, many reads are classified as chromosomal by the adaptive sampling tools and rejected from the pores, leading to more reads sequenced on the adaptive sampling regions. Here, the flow cell run ReadBouncer2 has a higher number of reads on the adaptive sampling region than ReadBouncer1. This results from a lower relative plasmid abundance in samples sequenced on flow cell ReadBouncer2, which leads to a larger number of chromosomal reads that were rejected on the adaptive sampling region of flow cell ReadBouncer2 (approx. 400,000) than on the adaptive sampling region of ReadBouncer1 (approx. 370,000)."

Line 156-157: The authors suggest that the used barcodes, among other reasons, might influence the obtained read lengths. This is not clear to me (as a computational biologist) and might benefit from additional clarification.

Since we can not justify the statement with an in-depth analysis, we decided to remove the statement from the results section, as suggested by reviewer #1.

Fig. 5: Both panels of results for ReadBouncer show two distinct densities for read lengths of the rejected DNA fragments. I could not find any discussion of this observation in the text or caption and am wondering if the authors have an explanation or hypothesis about this?

We agree with the reviewer and added Figure S12, showing the read length histograms of rejected reads separated by adaptive sampling tool. We further added the following statement to explain the issue in the results section:

"An analysis of rejected reads revealed shorter read lengths for MinKNOW compared to ReadBouncer, which is caused by rejection decisions based on short read prefixes (see Figure S12). In the histogram, we see that reads rejected by ReadBouncer are longer than those rejected by MinKNOW, with an average length of 848 bp compared to 520 bp. This confirmed our assumption that ReadBouncer rejects reads later during the adaptive sampling process, resulting in a higher abundance of unwanted chromosomal base pairs in the final output. To avoid confusion, we have to note that the lengths of rejected reads in the final output are not the same as the read prefix (or chunk) length used by adaptive sampling tools for making rejection decisions. Lengths of rejected reads in the final output represent the time needed for the whole decision process, including time for communication with the API and mapping of reads against index data structures."

Fig. 5: The x-axis labels indicate \log_{10} , when the presented values seem to not be log-values but actual read lengths.

While the x-axis in the mentioned figure is log-scaled, we agree that the labeling can be misleading. Thus, we removed the “log₁₀” labeling and mention that the axis is log_scaled. The figure itself has also been moved to the Supplemental Material to shorten the verbose beginning of the results section.

Lines 205-206: Here, the authors argue that MinKNOW achieves higher levels of enrichment due to faster rejection decisions. It could be helpful to be more precise in this statement, i.e. is the higher enrichment due to using less sequence data for alignment, is it due to differences in speed of performing the alignment, or perhaps due to other factors influencing the processing of read rejections?

In principle, the higher potential enrichment level is a result of the faster decision-making process by MinKNOW, which we showed by shorter read lengths of rejected reads in Figure S12 (see below)

Fig. 7: Panels b and d are comparisons of observed and predicted values. To ease the interpretation of the data, it would help to either make sure the panels have a square aspect ratio, to add a diagonal line to the plot, or both. In panels a and c, a horizontal line could be added to indicate the ratio of 1. Changing the y-axis range in panel c would also help the

reader in discerning differences and interpreting results. The caption text of panel c contains the same description as panel a. This redundant text could be replaced with more informative text that aids in understanding the results.

We appreciate the reviewer's comments and followed the suggestion by adding diagonal lines to panels b) and d) and horizontal lines at y-axis value 1 to panels a) and c). We further removed the redundant text from the figure caption and added some explanations to the captions for better understanding.

Line 208ff: The authors state that the original plasmid abundance does not impact the enrichment by yield, only the enrichment by composition. Could an explanation of this be added? Is this due to the difference of read lengths from chromosomal material and plasmid sequencing reads? It would in general be interesting to see the read length distribution separated by plasmid and chromosomal DNA; or whether there is any difference given the plasmid sizes (not mentioned in the manuscript, see above).

Following the reviewer's recommendations, we added read-length histograms separated by plasmid and chromosome and by control and adaptive sampling regions for each sample. Figures S5-S8 are part of the Supplemental Material, and one of them is provided below. We further added the following statement explaining the impact of plasmid abundance on enrichment:

“When using adaptive sampling, the composition of a sample changes from e.g. 90% chromosome/10% plasmid to 80% plasmid/20% chromosome. Here, we see that the lower the plasmid abundance was in the original sample, the higher is the fold change for this compositional abundance. However, plasmid abundance does not impact the fold change in sequenced plasmid bases when we use adaptive sampling. Irrespective of whether we had 5% or 10% plasmid bases in our sample, the relative enrichment in plasmid bases will be between 1.1 and 1.8.”

Lines 216ff: When referring to equations it would be helpful to reference the methods section in which they appear, especially since the methods section is printed after the results in this version of the manuscript.

We agree with the reviewer and changed the referencing of the equations by explicitly mentioning that they are presented in the methods section.

Lines 219, 223: Panel b of Fig. 8 is referenced before panel a. It might be worth swapping the figure panels if that is more natural to how the results are presented in the text.

We followed the reviewer's suggestion and swapped the figure panels. As the reviewer also suggested, We added a horizontal line at y-axis value 1 for the enrichment vs. abundance plots.

Fig. 8: As in the previous figure, it might be helpful to add a line at the ratio of 1. Additionally, it would be interesting to have an interpretation of the shape of the results shown here, i.e. the steep increase in the beginning, followed by a steady decline of the advantage as the experiment progresses. Is this related to a possibly increased pore degradation, depletion of plasmid sequence material, or any other factors?

We added the line at the ratio of 1 (see above) and the following statement in the results section:

“In general, we see a steep increase in enrichment at the beginning of each experiment, followed by a steady decline as the experiments progress. While the steep increase at the beginning is very surprising, we assume that the slow decrease is caused by pore degradation and most chromosomal reads being rejected early in the experiments, resulting in a relatively constant number of plasmid reads sequenced throughout the later stages in the experiments on both sides of the flow cells.”

Line 271: The authors state that adaptive sampling improved plasmid assemblies "in all cases". The results for *Salmonella enterica* of the left MinKNOW column, however, indicate that the control assembly of the plasmid is in fact better than the adaptive sampling assembly.

Besides adding assembly statistics for all sequenced samples as recommended by reviewer #1, we revisited the whole subsection that describes the assembly results and thus also changed this statement.

Line 402: I assume the references used for adaptive sampling correspond to the accessions indicated in the section "Culture and DNA extraction"? It would be helpful to clarify.

As also suggested by reviewer #1, we added more information about the used references. This also includes the reference accessions provided in Supplemental Table 1, which we refer to at the beginning of the results section. We also mention using the chromosomal references as depletion targets for adaptive sampling by adding the following statement: “The chromosome sizes of the five bacterial strains range from 1.6 to 5.4 Mb. We used these chromosomal references as depletion targets for all adaptive sampling experiments conducted in this study.”

Line 417: Raw data collected in 0.4s were used for real-time basecalling for both methods. Given the sequencing speed of 420 nucleotides/s stated in the manuscript, this corresponds to ~168nt. On lines 343-344 the authors discuss that unique sequences to discern similar segments would need to be contained in the first 1000nt. If only ~168nt are used to make decisions, would unique sequences not have to be present in that amount of data?

The reviewer is correct that 0.4s of sequencing correspond to ~168nt. However, the manuscript statement means that the ReadUntil API waits at most 0.4 seconds before sending a chunk of data. In fact, MinKNOW and ReadBouncer use more than 168nt at the

beginning of each fragment. They concatenate the basecalled sequences of the different data chunks from the same read before the decision-making algorithm decides which reads to reject. For MinKNOW, that parameter for the maximum number of data chunks cannot be tweaked. For ReadBouncer, this can be set. In general, MinKNOW uses approx. 300nt and ReadBouncer 500 to 700nt of the read prefix for decision-making. With regard to this point, we added several statements throughout the manuscript. One about the different read lengths used was already given above. In the discussion, we added the following:

“..the fact that ReadBouncer uses longer read prefixes for the decision-making algorithm seems to prevent false rejection decisions. Unfortunately, the prefix length, alignment identity or minimum alignment length for decision-making cannot be parameterized via MinKNOW, which suggests that more tunable tools like ReadBouncer are better suited for complex samples. However, for less complex samples MinKNOW is more user-friendly and potentially achieves higher enrichment values through faster rejection decisions.”

We further added a brief explanation of the “*break_reads_after_seconds*” parameter in the methods section:

“We set the *break_reads_after_seconds* parameter to 0.4, which results in receiving the first chunks of raw data from a read after 0.4 seconds. Both methods concatenate basecalled read chunks to longer prefixes if the prefixes are too short to reliably classify them as plasmid or chromosome.”

Re: mSystems00945-23R1 (Nanopore adaptive sampling effectively enriches bacterial plasmids)

Dear Mr. Jens-Uwe Ulrich:

Your manuscript has been accepted, and I am forwarding it to the ASM production staff for publication. Your paper will first be checked to make sure all elements meet the technical requirements. ASM staff will contact you if anything needs to be revised before copyediting and production can begin. Otherwise, you will be notified when your proofs are ready to be viewed.

Featured Image Submissions: If you would like to submit a potential Featured Image, please email a file and a short legend to msystems@asmusa.org. Please note that we can only consider images that (i) the authors created or own and (ii) have not been previously published. By submitting, you agree that the image can be used under the same terms as the published article. Image File requirements: TIF/EPS, 7.5 inches wide by 8.25 inches tall (at least 2,250 pixels wide by 2,475 pixels tall), minimum 300 dpi resolution (600 dpi preferred), RGB, and no figure elements, e.g., arrows or panel labels. The legend should be a short description of the image, 1-2 sentences recommended.

Sincerely,
Juliette Hayer
Editor

mSystems

Reviewer #1 (Comments for the Author):

The authors have comprehensively addressed our comments! Thank you!

Reviewer #2 (Comments for the Author):

The authors have thoughtfully and adequately addressed my questions in detail. The addition of several supplementary figures and modifications of previous figures are useful updates. I, and hopefully future readers, especially appreciate the clarification of differences between ReadBouncer and adaptive sampling with MinKNOW, e.g. the differences in the length of read-prefixes used for decision-making. Overall, I believe the authors have addressed all comments appropriately.